# Higher-order Granger reservoir computing: simultaneously achieving scalable complex structures inference and accurate dynamics prediction

Xin Li[1,2], Qunxi Zhu [2,3] ✉, Chengli Zhao [1] ✉, Xiaojun Duan[1], Bolin Zhao[2,3], Xue Zhang[1], Huanfei Ma [4], Jie Sun[2,5] & Wei Lin [2,3,6] ✉

Recently, machine learning methods, including reservoir computing (RC), have been tremendously successful in predicting complex dynamics in many fields. However, a present challenge lies in pushing for the limit of prediction accuracy while maintaining the low complexity of the model. Here, we design a data-driven, model-free framework named higher-order Granger reservoir computing (HoGRC), which owns two major missions: The first is to infer the higher-order structures incorporating the idea of Granger causality with the RC, and, simultaneously, the second is to realize multi-step prediction by feeding the time series and the inferred higher-order information into HoGRC. We demonstrate the efficacy and robustness of the HoGRC using several representative systems, including the classical chaotic systems, the network dynamical systems, and the UK power grid system. In the era of machine learning and complex systems, we anticipate a broad application of the HoGRC framework in structure inference and dynamics prediction.

Machine learning has been recently recognized as a vital engine in efficiently addressing numerous scientific and real-world problems that are not easily solvable using traditional methods[1–6]. To this end, a significant effort has been devoted to applying model-free, machine learning methods to those observational data of time series for analyzing and predicting complex dynamics, attracting tremendous attention[7–13]. Despite initial or/and partial successes, those machine learning methods still meet difficulties in typical scenarios where the investigated complex systems are of higher dimensions, replete with different types of interactions, and even exhibiting highly complex dynamical behaviors[14–18]. Thus, it is crucial to develop and implement delicate machine learning methods for not only uncovering internal interactions in such complex systems but also predicting their future evolution by leveraging the discovered interactions.

Compared to classical methods such as auto-regressive models (ARMA)[19] and multi-layer perceptron (MLP)[20], machine learning techniques such as the recurrent neural networks (RNNs)[21], neural ordinary differential equations (NODEs)[22], and deep residual learning[23] offer several advantages for analyzing time series data generated by nonlinear and complex systems. Specifically, RNNs and their variants, including long short-term memory (LSTM)[24] networks and gated recurrent units (GRU)[25], exhibit excellent performance in predicting dynamics but require estimation of many parameters. In addition to

[1]Center for Applied Mathematics (NUDT), Changsha 410073 Hunan, China. [2]Research Institute of Intelligent Complex Systems and MOE Frontiers Center for Brain Science, Fudan University, Shanghai 200433, China. [3]School of Mathematical Sciences, SCMS, SCAM, and CCSB, Fudan University, Shanghai 200433, China. [4]School of Mathematical Sciences, Soochow University, Suzhou 215006, China. [5]HUAWEI Technologies Co., Ltd., Hong Kong, China. [6]Shanghai Artificial Intelligence Laboratory, Shanghai 200232, China. ✉e-mail: qxzhu16@fudan.edu.cn; chenglizhao@foxmail.com; wlin@fudan.edu.cn

these networks with a huge number of parameters for updating, reservoir computing (RC), a lightweight RNN, was recently proposed for predicting temporal-spatial behaviors of chaotic dynamics and aroused great interest[26–31]. Actually, in an RC, the hidden states are of high dimension and only the weights of the output layer require training. As a result, it possesses a strong modeling ability but needs less computational cost.

Although the advantages of the RC framework have been validated in many scenarios[32–34], there is still room for improvement so that outstanding endeavors have been paid for recently and persistently. Examples abound: Lu and Lukoševičius et al. added nonlinear terms of hidden states and raw data, respectively, in the output layer to enhance the modeling ability of the RC[35,36]; Gauthier et al. introduced some nonlinear combinations of the original data into the input layer to greatly improve the computational efficiency[37], and Gallicchio et al. extended the RC to its deep network forms[38]. While these approaches improve the performance of RC, they encounter difficulties when the dynamics dimension is higher, the nonlinearity is stronger and the structure is more complex. To exceed the ceiling, the latest works in refs. [39,40] proposed a parallel forecasting method, parallel RC (PRC), for complex dynamical networks, using the local structure of systems. These pairwise structures used in the PRC method can be obtained through traditional causal inference methods and their improved variants[41–46]; however, they cannot uncover directly the higher-order structures, a kind of more complex interactions that are ubiquitous in complex dynamical systems. In fact, recent studies show that the higher-order structures are vital to the emergence of complex dynamics[47], viz. diffusion[48], synchronization[49], and evolutionary processes[50]. It thus is believed that an appropriate introduction of not only the traditional structural information but also the higher-order structures into the RC is beneficial to achieving more accurate and long-term predictions. In addition, conventional system identification algorithms including SINDy (sparse identification of the nonlinear dynamics)[16,51,52] or entropic regression[53] aim to fit equations using a predefined set of basis functions in dynamical systems. However, these methods have certain limitations. They are restricted to a particular set of bases and necessitate high-quality observational data. When there are more complex interactions within the system, the risk of producing an erroneous sparse model increases. Such incorrect identification of interactions may inevitably lead to catastrophic predictive performance, while a simple RC even without any structure information can often yield satisfactory results. Naturally and consequently, two missions are at hand: 1) the inference of higher-order structures solely based on observational data, and 2) the utilization of the inferred optimal structures to make more accurate and long-term predictions.

To address the aforementioned issue, we propose a novel computing paradigm called higher-order RC, which aims to embed structural information, especially the higher-order structures, into the reservoir. However, the higher-order structures of the underlying complex dynamical systems are commonly unknown a priori. To this end, we incorporate the concept of Granger causality (GC) into the higher-order RC to identify the system's underlying higher-order interactions in an iterative manner, thereby enabling more accurate dynamical predictions with the inferred optimal higher-order structures. During this process, GC inference and RC prediction are performed *simultaneously* and complement each other, hence named as Higher-Order Granger RC (HoGRC) framework. This framework is highly scalable, in that, at the node level, simultaneously achieved are complex structure inference and accurate dynamics prediction. This therefore makes the developed framework applicable widely to higher-dimensional and more intricate dynamical systems.

## Results

### Classical reservoir computing

We start with a nonlinear dynamical network of $N$ variables of the following general form,

$$\dot{\mathbf{x}}(t) = \boldsymbol{f}[\mathbf{x}(t)], \quad (1)$$

where $\mathbf{x}(t) = [x_1(t), \ldots, x_N(t)]^\top$ denotes the $N$-dimensional ($N$-D) state of the system at time $t$, and $\boldsymbol{f}[\mathbf{x}(t)] = (f_1[\mathbf{x}(t)], f_2[\mathbf{x}(t)], \ldots, f_N[\mathbf{x}(t)])^\top$ is the $N$-D nonlinear vector field. In this article, we assume that neither the vector field $\boldsymbol{f}$ (equivalently, each element $f_i$) nor the underlying complex interaction mechanism among these $N$ variables is partially or completely unknown a prior. The only available information about the underlying system is the observational time series $\mathbf{x}(t)$ at the discrete time steps. Here, we choose a regularly sampled time increment $\Delta t$.

The traditional RC, a powerful tool for modeling time series data, embeds the observational data $\mathbf{x}(t)$ into an $n$-dimensional hidden state $\mathbf{r}(t)$ using an input matrix $\mathbf{W}_{in}$ of dimension $n \times N$. Then the hidden state $\mathbf{r}(t)$ evolves within the reservoir with a weighted adjacency matrix $\mathbf{A}$ of dimension $n \times n$, given by

$$\mathbf{r}(t + \Delta t) = (1 - l) \cdot \mathbf{r}(t) + l \cdot \tanh\left[\mathbf{W}_{in}\mathbf{x}(t) + \mathbf{A}\mathbf{r}(t) + \mathbf{b}_r\right], \quad (2)$$

where $l$ is the leaky rate and $\mathbf{b}_r$ is the bias term. Subsequently, an additional output layer is employed, typically implemented as a simple linear transformation using the matrix $\mathbf{W}_{out}$, mapping the reservoir state space to the desired output space. Here, the output space is the original data space,

$$\hat{\mathbf{x}}(t + \Delta t) = \mathbf{x}(t) + \mathbf{W}_{out}\mathbf{r}(t + \Delta t), \quad (3)$$

where $\mathbf{W}_{out}\mathbf{r}(t + \Delta t)$ can be explained as the predicted residue between $\mathbf{x}(t + \Delta t)$ and $\mathbf{x}(t)$, or equivalently the approximated integral operator $\int_t^{t + \Delta t} \boldsymbol{f}[\mathbf{x}(\tau)]\mathrm{d}\tau$. It is important to note that the only trained module is the output layer, i.e., $\mathbf{W}_{out}$, which can be solved explicitly via the Tikhonov regularized regression[54] with the loss unction:

$$\mathcal{L}_{\Delta t} = \sum_t \left\{ \mathbf{W}_{out}\mathbf{r}(t + \Delta t) - [\mathbf{x}(t + \Delta t) - \mathbf{x}(t)] \right\}^2 + \lambda_W \cdot \| \mathbf{W}_{out} \|, \quad (4)$$

where $\lambda_W$ is the regularization coefficient. By leveraging the trained RC, one can accurately achieve dynamics prediction.

### Higher-order structure in dynamical systems

To establish our framework, we first introduce a few important definitions about the higher-order structure for any given function of vector field based on the simplicial complexes summarized in[55].

**Definition 1. Separable and inseparable functions.** Assume that $g(\mathbf{s})$ is an arbitrarily given scalar function with respect to $\mathbf{s} = \{v_1, v_2, \ldots, v_k\}$, a non-empty set containing $k$ variables. If there are two variable sets $\mathbf{s}_1, \mathbf{s}_2 \in \{\mathbf{s}_1, \mathbf{s}_2 | \mathbf{s}_1 \not\subset \mathbf{s}_2, \mathbf{s}_2 \not\subset \mathbf{s}_1, \mathbf{s}_1 \cup \mathbf{s}_2 = \mathbf{s}\}$, and two scalar functions $g_1$ and $g_2$ such that

$$g(\mathbf{s}) = g_1(\mathbf{s}_1) + g_2(\mathbf{s}_2), \quad (5)$$

then $g(\mathbf{s})$ is a separable function with respect to $\mathbf{s}$, i.e., $g(\mathbf{s})$ can be decomposed into the sum of two functions whose variable sets have no inclusion relationship; otherwise $g(\mathbf{s})$ is an inseparable function.

**Definition 2. Higher-order neighbors.** Consider the nonlinear scalar differential equation $\dot{u} = g(\mathbf{s}_u)$, where $g(\mathbf{s}_u)$ is a scalar function with respect to a set of variables $\mathbf{s}_u$. We decompose the function $g(\mathbf{s}_u)$ into a

sum of several inseparable functions $g_i(\mathbf{s}_{u,i})$ as

$$g(\mathbf{s}_u) = g_1(\mathbf{s}_{u,1}) + g_2(\mathbf{s}_{u,2}) + \ldots + g_{D_u}(\mathbf{s}_{u,D_u}),$$
$$\mathbf{s}_u = \mathbf{s}_{u,1} \cup \mathbf{s}_{u,2} \cup \cdots \cup \mathbf{s}_{u,D_u}, \ \mathbf{s}_{u,i} \not\subset \mathbf{s}_{u,j}, \tag{6}$$

for all $i, j \in \{1, 2, \ldots, D_u\}$ with $i \neq j$, where $D_u$ is the number of terms. Then, we name the set $\mathbf{s}_{u,i} = \{v_{i_1}, v_{i_2}, \ldots, v_{i_{k_i}}\}$ as the $(k_i{-}1)$-D simplicial complex, and the $i$-th higher-order neighbor of node $u$. Denote by $\mathscr{S}_u = \{\mathbf{s}_{u,1}, \mathbf{s}_{u,2}, \ldots, \mathbf{s}_{u,D_u}\}$ the set of the higher-order neighbors of node $u$. We construct a hypergraph or a hypernetwork, denoted by $\mathcal{G} = (V, S)$, of system (1) under consideration. Here, $V = \{x_1, x_2, \ldots, x_N\}$ denotes the set of nodes, corresponding to the state variables of the system. According to Definitions 1 & 2, we introduce the concept of the higher-order neighbors $\mathscr{S}_u$ of an arbitrary node $u \in V$, yielding the set of higher-order neighbors for all nodes $S = \{\mathscr{S}_{x_1}, \mathscr{S}_{x_2}, \ldots, \mathscr{S}_{x_N}\}$. Hereafter, for simplicity of notation's usage, node $u$ is used as a placeholder of any element in the set $V$.

To better elucidate these concepts, we directly utilize the Lorenz63 system as an illustrative example. As shown in "Explanation (1)" of Fig. 1, for the third node $u = z$ in system (12), we write out

$$\dot{z} = f_3(x, y, z) = -\beta z + xy = g(x, y, z) = g_1(z) + g_2(x, y), \tag{7}$$

where $g_1(z) \triangleq -\beta z$, $g_2(x, y) \triangleq xy$, and $D_z \triangleq 2$. Consequently, according to Definitions 1 & 2, the set of the higher-order neighbors of node $u = z$ is $\mathscr{S}_z = \{\mathbf{s}_{z,1}, \mathbf{s}_{z,2}\} = \{\{z\}, \{x, y\}\}$. Similarly, we have $\mathscr{S}_x = \{\mathbf{s}_{x,1}, \mathbf{s}_{x,2}\} = \{\{x\}, \{y\}\}$ for node $u = x$ and $\mathscr{S}_y = \{\mathbf{s}_{y,1}, \mathbf{s}_{y,2}\} = \{\{y\}, \{x, z\}\}$ for node $u = y$. Consequently, we obtain the higher-order structure of the Lorenz63 system as $\mathcal{G} = (V, S) = ((x, y, z), (\mathscr{S}_x, \mathscr{S}_y, \mathscr{S}_z))$.

## A paradigm of reservoir computing with structure input

Despite the tremendous success achieved by the traditional RC in dynamics predictions in many fields, a difficulty still lies in pushing for the limit of prediction accuracy while maintaining the low complexity of the model. We attribute this difficulty to a lack of direct utilization of the structural information from the underlying dynamical system, since the structure is an important component of the system. Actually, the PRC, the recent framework[40] integrated pairwise structures to predict dynamics in complex systems. However, they cannot reveal the higher-order structures, a more precise representation of the complex interactions in complex dynamical systems.

Thus, we introduce a new computing paradigm into the RC, termed higher-order RC, to incorporate the time-series data with the higher-order structure to make accurate dynamics predictions. Specifically, as shown in Fig. 1b, we model each state variable (i.e., node $u$, as defined above) of the original system independently with a block of $n$ neurons in a reservoir network. Then we incorporate the higher-order neighbors of node $u$ into the corresponding RC, defined as $\mathcal{R}_u$. Subsequently, inspired by but different from the classical RC method (2), the hidden dynamics in the higher-order $\mathcal{R}_u$

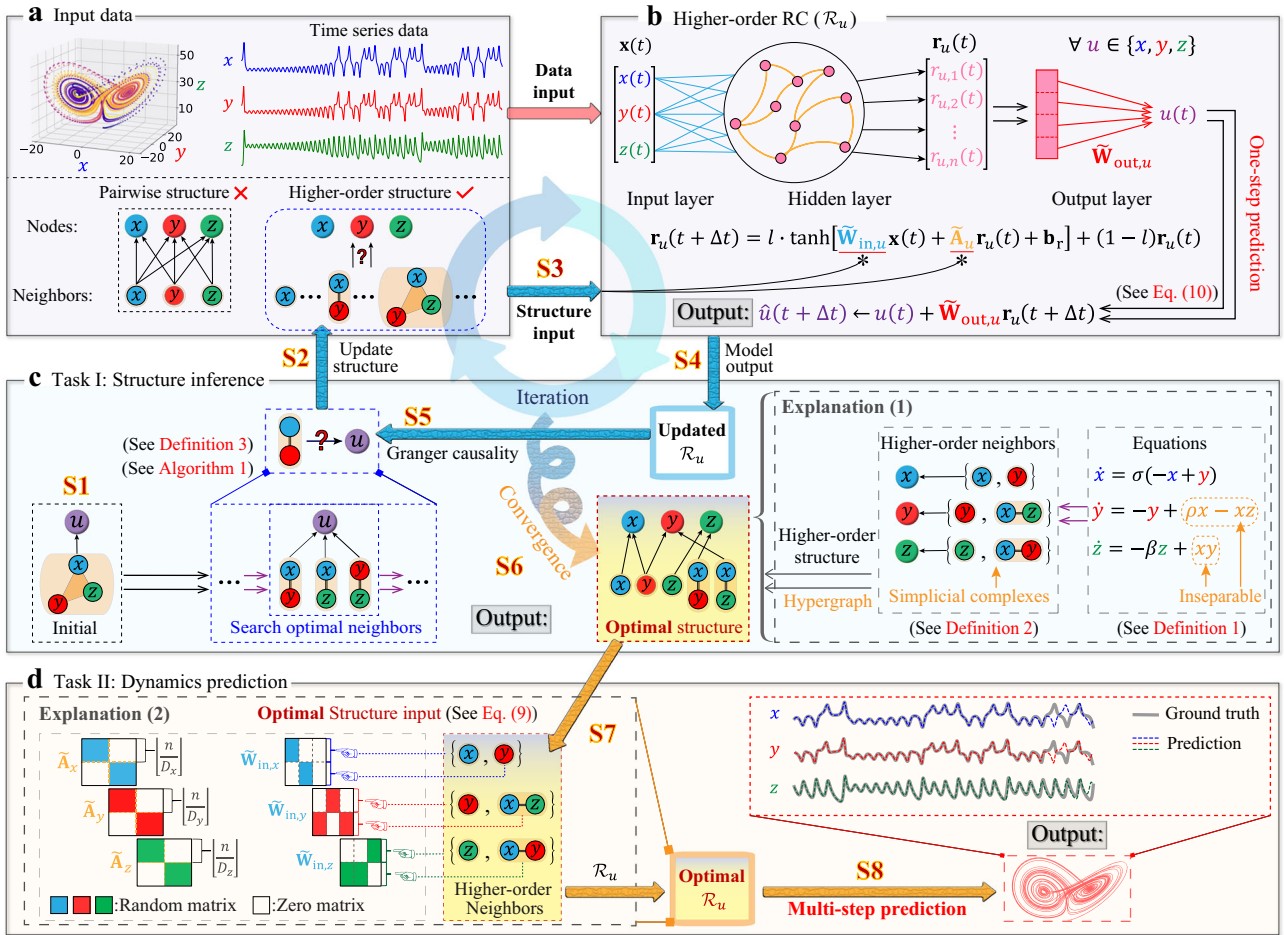

**Fig. 1 | Schematic diagrams for illustrating the proposed HoGRC framework.**
**a** The input data consists of time series data and higher-order structure information. **b** The new paradigm $\mathcal{R}_u$ using the higher-order structural information. **c** The HoGRC framework enables the inference of higher-order structures. **d** The HoGRC framework achieves multi-step dynamics prediction using the inferred optimal structure. The markers "S1"–"S8" correspond to the steps in Table 2. We offer "Explanation 1" to elucidate the concept of higher-order structures, and use "Explanation 2" to clarify the notion of the higher-order structure embedding. Due to the same process and the independently trained $\mathcal{R}_u$ for all nodes, it makes the HoGRC own the scalability or parallel merit[40].

is given by

$$\mathbf{r}_u(t + \Delta t) = (1 - l) \cdot \mathbf{r}_u(t) + l \cdot \tanh\left[\tilde{\mathbf{W}}_{\text{in},u}\mathbf{x}(t) + \tilde{\mathbf{A}}_u\mathbf{r}_u(t) + \mathbf{b}_r\right], \quad (8)$$

for different $u \in V$. Thus, we establish a total of $|V|$ sub-RC networks, where $|V|$ denotes the number of the elements in the set $V$. In contrast to the traditional RC method (2) that solely relies on a single random matrix $\mathbf{W}_{\text{in}}$ and a single random matrix $\mathbf{A}$ without including any higher-order structural information, the framework (8) operates at node level, notably incorporating the corresponding higher-order structural information. Specifically for each node $u \in V$, this framework embeds the higher-order structural information directly into the matrices $\tilde{\mathbf{W}}_{\text{in},u}$ and $\tilde{\mathbf{A}}_u$ in the following forms:

$$\tilde{\mathbf{W}}_{\text{in},u} = \left[\psi^\top(\mathbf{s}_{u,1}), \psi^\top(\mathbf{s}_{u,2}), \dots, \psi^\top(\mathbf{s}_{u,D_u})\right]^\top \in \mathbb{R}^{n \times N}, \quad \psi(\mathbf{s}_{u,i}) \in \mathbb{R}^{\lfloor n/D_u \rfloor \times N},$$
$$\tilde{\mathbf{A}}_u = \text{diag}\{\varphi(\mathbf{s}_{u,1}), \varphi(\mathbf{s}_{u,2}), \dots, \varphi(\mathbf{s}_{u,D_u})\} \in \mathbb{R}^{n \times n}, \quad \varphi(\mathbf{s}_{u,i}) \in \mathbb{R}^{\lfloor n/D_u \rfloor \times \lfloor n/D_u \rfloor}, \quad (9)$$

where $\lfloor \cdot \rfloor$ is the floor function, and the integer $n$ is selected as a multiple of $D_u$. Different from $\mathbf{W}_{\text{in}}$, a randomly initialized matrix in its entirety, in the traditional RC framework (2), each $\psi(\mathbf{s}_{u,i})$ in $\tilde{\mathbf{W}}_{\text{in},u}$ is a random (resp., zero) block submatrix of dimension $\lfloor n/D_u \rfloor \times N$ such that, if $x_j \in$ (resp., $\notin$) $\mathbf{s}_{u,i}$ for $j \in \{1, 2, \dots, N\}$, all elements of the $j$-th column of $\psi(\mathbf{s}_{u,i})$ are set as random values (resp., zeros), and $\varphi(\mathbf{s}_{u,i})$ represents a random sparse submatrix of dimension $\lfloor n/D_u \rfloor \times \lfloor n/D_u \rfloor$. Actually, these block configurations in the reservoir facilitate a more precise utilization of the higher-order structural information.

To enhance the transparency of the above configurations, we provide a visual representation in "Explanation (2)" of Fig. 1, where depicted is the true higher-order RC structure (i.e., the optimal network finally obtained in the following inference task, see the next subsection) under consideration of the Lorenz63 system. Specifically, as mentioned above, for node $u = z$, the set of the higher-order neighbors becomes $\{\{z\}, \{x, y\}\}$ with $D_z = 2$. Thus, we obtain $\tilde{\mathbf{W}}_{\text{in},z} = [\psi^\top(z), \psi^\top((x,y))]^\top$ according to the notations set in (9), where the third column of $\psi^\top(z)$ and the first and the second columns of $\psi^\top[(x,y)]$ are the random sparse submatrices, and the remaining parts are zero submatrices. Moreover, we obtain $\tilde{\mathbf{A}}_z = \text{diag}\{\varphi(z), \varphi[(x,y)]\}$, which is a block diagonal matrix comprising two random sparse submatrices. Additionally, we provide a simple illustrative example about the difference between the traditional RC method (2) and the newly proposed higher-order RC framework (8) in Supplementary Note 1.3.

Now, by embedding the higher-order structural information into the dynamics of the reservoir in the above manner, we obtain the $n$-D hidden state $\mathbf{r}_u(t) = [r_{u,1}, r_{u,2}, \dots, r_{u,n}]^\top(t)$ for each $u \in V$. This allows us to predict the system's state $u$ in the next time step as

$$\hat{u}(t + \Delta t) = u(t) + \tilde{\mathbf{W}}_{\text{out},u}\mathbf{r}_u(t + \Delta t), \quad (10)$$

where $\tilde{\mathbf{W}}_{\text{out},u}$ represents an output matrix of dimension $1 \times n$, employed for the prediction of $u$.

Significantly, our framework fully inherits the parallel merit of the existing work[40]. In particular, the above process operates at the node level, focusing exclusively on every node $u$, and such a process can be applied across all nodes in $V$. Different from the classical RC, we use a specific higher-order $\mathcal{R}_u$ to model each node $u$, thereby requiring a smaller reservoir size $n$ or resulting in a lightweight model. Moreover, since all lightweight reservoirs $\mathcal{R}_u$ ($u \in V$) are independently trained, our framework can be efficiently processed in a parallel manner, which in turn makes our framework *scalable* to higher-dimensional systems.

## Integration of structure inference and dynamics prediction

In the preceding section, the setup of the higher-order $\mathcal{R}_u$ requires the exact information of the structures. However, in real-world scenarios, the specific form as well as the higher-order structures of a system are always unknown before the setup of $\mathcal{R}_u$. So, we design an iterative algorithm to seek the optimal structure for $\mathcal{R}_u$ which is initially endowed with a structure containing all possible candidates or only partially known information. To carry out this design, we novelly integrate the concept of the Granger causality (GC) into the higher-oder $\mathcal{R}_u$ (see Table 1). Subsequently, the inferred structures are utilized to update $\mathcal{R}_u$, thereby further enhancing its prediction performance. This iterative procedure is repeated until the model achieves optimal prediction accuracy. Consequently, we refer to this integrated model as the HoGRC framework, as depicted in the composite of Fig. 1a-d.

Particularly, we develop an efficient greedy strategy, as outlined in Table 1 of the Methods section, to infer the true higher-order structure of system (1) solely from the time series data. As shown in Fig. 1c, for any node $u$, we employ the one-step prediction error of $\mathcal{R}_u$ based on the concept of the GC (see Definition 3) to iteratively refine the initial and coarse-grained candidate neighbors into the optimal and fine-grained higher-order neighbors, until an optimal structure is obtained, tending to align with the true higher-order structure defined in Definition 2. In the iterative procedure, the GC inference and the dynamics prediction using $\mathcal{R}_u$ are complementarily and mutually reinforcing. As depicted by the blue loop in Fig. 1, the structure discovered by the GC significantly enhances the predictability of $\mathcal{R}_u$, and conversely, the updated $\mathcal{R}_u$ in the iterative procedure makes the GC discover the structure in a more effective manner.

Furthermore, as indicated by the orange arrows in Fig. 1d, we obtain the optimal $\mathcal{R}_u$ for all nodes $u$ based on the input of the optimal higher-order structure. Then, these optimal models can perform multi-step prediction by continually adding the most recent forecasted values to the input data, which significantly outperforms the traditional prediction methods. Therefore, the HoGRC framework, integrating the node-level RC and the GC inference, simultaneously achieve two functions: (I) structures inference (Fig. 1c) and (II) dynamics prediction (Fig. 1d). To enhance comprehension of the HoGRC workflow, we provide a summary of the key execution steps in Table 2, where the steps correspond to the markers "S1"–"S8" in Fig. 1.

## Table 1 | The process of inferring higher-order neighbors using Algorithm 1

| Algorithm 1: Inferring higher-order neighbors. |
|---|
| **Data:** Set the initial candidate neighbor set of node $u$ as $\mathscr{C}_0 = \{\mathbf{c}_{1,0}, \dots, \mathbf{c}_{K_0,0}\}$, and denote by $\mathbf{x}(t) = (x_1(t), x_2(t), \dots, x_N(t))$ the time series data. |
| **Result:** The higher-order neighbors of node $u$ are inferred as $\mathscr{S}_u = \{\mathbf{s}_{u,1}, \dots, \mathbf{s}_{u,D_u}\}$. |
| **Step1:** Set a suitable threshold $\epsilon_e$, and let $\mathscr{C} = \mathscr{C}_0$; |
| **Step2:** Traverse through all the possible complex in $\mathscr{C}$, and delete any non-causal factor that satisfies Definition 3; |
| **Step3:** Rearrange the elements in $\mathscr{C}$ from high order to low order; |
| **Step4:** Reduce the dimensionality of the complex $\mathscr{C}$. Traverse through all the possible complex $\mathbf{c}$ in $\mathscr{C}$. If $\mathbf{c}$ is a $(d_c + 1)$-D complex and $d_c \geq 0$, then try to reduce it to $(d_c + 2)$ complexes of dimension $d_c$ based on the error threshold $\epsilon_e$; |
| **Step5:** If $\mathscr{C}$ has not changed in this iteration, output $\mathscr{S}_u = \mathscr{C}$; otherwise, return to Step 2. |

**Table 2 | Main steps of the HoGRC framework**

| |
|---|
| **S1**: For any node (state variable) $u \in V$ in the considered system (1), set the initial candidate neighbors (possibly containing all or some nodes, depending on the knowledge about the system's structure, completely unknown or partially known). |
| **S2**: Update the higher-order structure of node $u$ using the results yielded by **S5** (using the initial candidate neighbors for the initial iteration). |
| **S3**: Input the updated structure and the temporal data into the higher-order $\mathcal{R}_u$. |
| **S4**: Train and update $\mathcal{R}_u$ using the updated structure and temporal data. |
| **S5**: Optimize the candidate neighbors of node $u$ based on the updated higher-order $\mathcal{R}_u$ and the *Granger-causality*-like rule (see Definition 3 and Algorithm 1 of Table 1 for details). |
| **S6**: If the higher-order neighbors of node $u$ no longer change, we obtain the optimal higher-order neighbors (Task I); otherwise, return to **S2**. Perform the above process (**S1**–**S6**) in a parallel manner for all nodes in $V$. |
| **S7**: Obtain $N$ independently optimal $\mathcal{R}_u$'s using the obtained optimal structure. |
| **S8**: Utilize these optimal $\mathcal{R}_u$'s to perform multi-step dynamics prediction (Task II). |

For more detailed information about the HoGRC framework, please refer to Methods section.

## Evaluation metrics

To demonstrate the efficacy of the two tasks achieved by the proposed framework, we conduct experiments using several representative systems from different fields. For Task (I), we utilize the one-step extrapolation prediction error produced by the HoGRC framework to search the higher-order neighbors of all dimensions in order to identify the higher-order structure with higher accuracy. For Task (II), we test the classical RC, the PRC[40], and the HoGRC, respectively, on several representative dynamical systems and compare their prediction performances (see Methods section for the differences among these three methods). For a clearer illustration, we define the valid predictive steps (VPS) as the predictive time steps when the prediction accuracy exceeds a certain threshold. Additionally, we adopt the root mean square error (RMSE) as a metric to quantitatively evaluate the prediction error,

$$\mathrm{RMSE}(t) = \sqrt{\frac{1}{N}\sum_{i=1}^{N}\left[\frac{\hat{x}_i(t)-x_i(t)}{\sigma_i}\right]^2}, \qquad (11)$$

where $\sigma_i$ is the standard deviation of $x_i(t)$. In our work, we use the VPS to evaluate the prediction performance of the HoGRC, i.e., $\mathrm{VPS} = \inf\{s : \mathrm{RMSE}(s\Delta t) > \epsilon_r\}$, where $\epsilon_r$ is the positive threshold and $\Delta t$ is the time step size. In the following numerical simulations, without a specific statement, we always set $\epsilon_r = 0.01$.

## Performances in representative dynamical systems

Here, we aim to demonstrate the effectiveness of the HoGRC framework using several representative dynamical systems. We take a 3-D Lorenz63 system and a 15-D coupled Lorenz63 system as examples. Additional experiments for more systems are included in Supplementary Note 2.

First, we consider the Lorenz63 system[56] which is a typical chaotic model described by the following equations:

$$\begin{aligned}
\dot{x} &= f_1(x,y,z) = \sigma(y-x), \\
\dot{y} &= f_2(x,y,z) = \rho x - y - xz, \\
\dot{z} &= f_3(x,y,z) = -\beta z + xy,
\end{aligned} \qquad (12)$$

where $\sigma, \beta, \rho$ are system parameters. In the simulations, we take the first 60% of the data generated by the system as the training set, and reserve the remaining data for testing purposes.

We begin our analysis by using the proposed method to identify the higher-order neighbors of the considered system. All the other hyperparameters of the RC, the PRC, and the HoGRC are specified, respectively, in Supplementary Note 3. Subsequently, we employ Algorithm 1 of Table 1 to infer the higher-order neighbors of all nodes

in the Lorenz63 system. Specifically, Fig. 2a presents an inference process for node $z$ using Algorithm 1 of Table 1, a greedy strategy. At the beginning, when no information regarding the network structure is available, the set of the higher-order neighbors for node $z$ is initially assigned as $\mathscr{C}_z = \mathscr{C}_z^0 = \{\{x,y,z\}\}$. Thus, $\tilde{\mathbf{W}}_{\mathrm{in},z}$ and $\tilde{\mathbf{A}}_z$, the input and the adjacency matrices, are constructed with $\mathscr{C}_z^0$, and $\mathcal{R}_z^0$, the corresponding higher-order RC, is utilized to calculate the one-step prediction error $e(z)$, designated as $e_1$. Next, one needs to decide whether to rectify $\mathscr{C}_z$ by reducing the dimensionality based on Algorithm 1 of Table 1. To do so, set $\mathscr{C}_z^1 = \{\{x,y\},\{y,z\},\{x,z\}\}$, and then the prediction error $e_2$ is obtained using $\mathcal{R}_z^1$ with $\mathscr{C}_z^1$. Here, by setting a small threshold $\epsilon_e$ (e.g., $10^{-7}$), it is found that $e_1 + \epsilon_e \geq e_2$, which implies a prediction promotion and thus, results in a resetting $\mathscr{C}_z = \mathscr{C}_z^1$ based on Definition 3. Then, one needs to decide whether to delete any element, e.g. $\{y,z\}$, in the current set $\mathscr{C}_z$. To do so, set $\mathscr{C}_z^2 = \{\{x,y\},\{x,z\}\}$. Thus, the prediction error $e_3$ is obtained using $\mathcal{R}_z^2$ with $\mathscr{C}_z^2$, which further yields $e_2 + \epsilon_e \geq e_3$. This prediction promotion leads us to reset $\mathscr{C}_z = \mathscr{C}_z^2$. However, as the sets $\mathscr{C}_z^3 = \{\{x,z\}\}$ and $\mathscr{C}_z^4 = \{\{x,z\}\}$ are, respectively, taken into account, $e_3 + \epsilon_e < e_4$ and $e_3 + \epsilon_e < e_5$ are obtained using $\mathcal{R}_z^3$ with $\mathscr{C}_z^3$ and $\mathcal{R}_z^4$ with $\mathscr{C}_z^4$, respectively. These inequalities indicate that there is no improvement in prediction and, consequently, no rectification needed for the set $\mathscr{C}_z$ at this stage. Therefore, the set should remain unaltered as $\mathscr{C}_z = \mathscr{C}_z^2$. In what follows, one still needs to decide whether to further rectify $\mathscr{C}_z$ by reducing the dimensionality based on Algorithm 1 of Table 1. To do so, set $\mathscr{C}_z^5 = \{\{x,y\},\{z\}\}$. Thus, $e_6$ and $e_3 + \epsilon_e \geq e_6$ are obtained, which leads us to further reset $\mathscr{C}_z = \mathscr{C}_z^5$. As suggested in Fig. 2, prediction is not improved by further reducing the dimensionality of $\mathscr{C}_z$ as $\mathscr{C}_z^6 = \{\{x\},\{y\},\{z\}\}$. This, with the greedy strategy we use, indicates an iteration terminal for inferring the higher-order neighbors with an output $\mathscr{S}_z = \mathscr{C} = \mathscr{C}_z^5$. Here, actually $\mathcal{R}_z^5$ with $\mathscr{S}_z = \mathscr{C}_z^5$ after training is the optimal higher-order RC of dynamics reconstruction and prediction for the state of node $z$. In addition, the inferred results of nodes $x$ and $y$ can be found in Supplementary Note 4.1.

In Task (II), we perform multi-step prediction using different methods, we find that the HoGRC framework yields the best prediction despite utilizing information solely from higher-order neighbors (see Supplementary Note 4.1). Additionally, in Supplementary Note 2, we also conduct similar experiments using other classic chaotic systems. Our findings indicate that systems with stronger nonlinearity and more complex structures tend to exhibit better prediction performance using the HoGRC framework.

Next, we investigate the coupled Lorenz63 (CL63) system[34] with a more complex structure and stronger nonlinear interactions, in which the dynamical behaviors of each subsystem is described by:

$$\begin{aligned}
\dot{x}_i &= -\sigma\left[x_i - y_i + \gamma\sum_{j=1}^{m} w_{ij}g_{ij}(y_i,y_j)\right], \\
\dot{y}_i &= \rho(1+h_i)x_i - y_i - x_iz_i, \quad \dot{z}_i = x_iy_i - \beta z_i,
\end{aligned} \qquad (13)$$

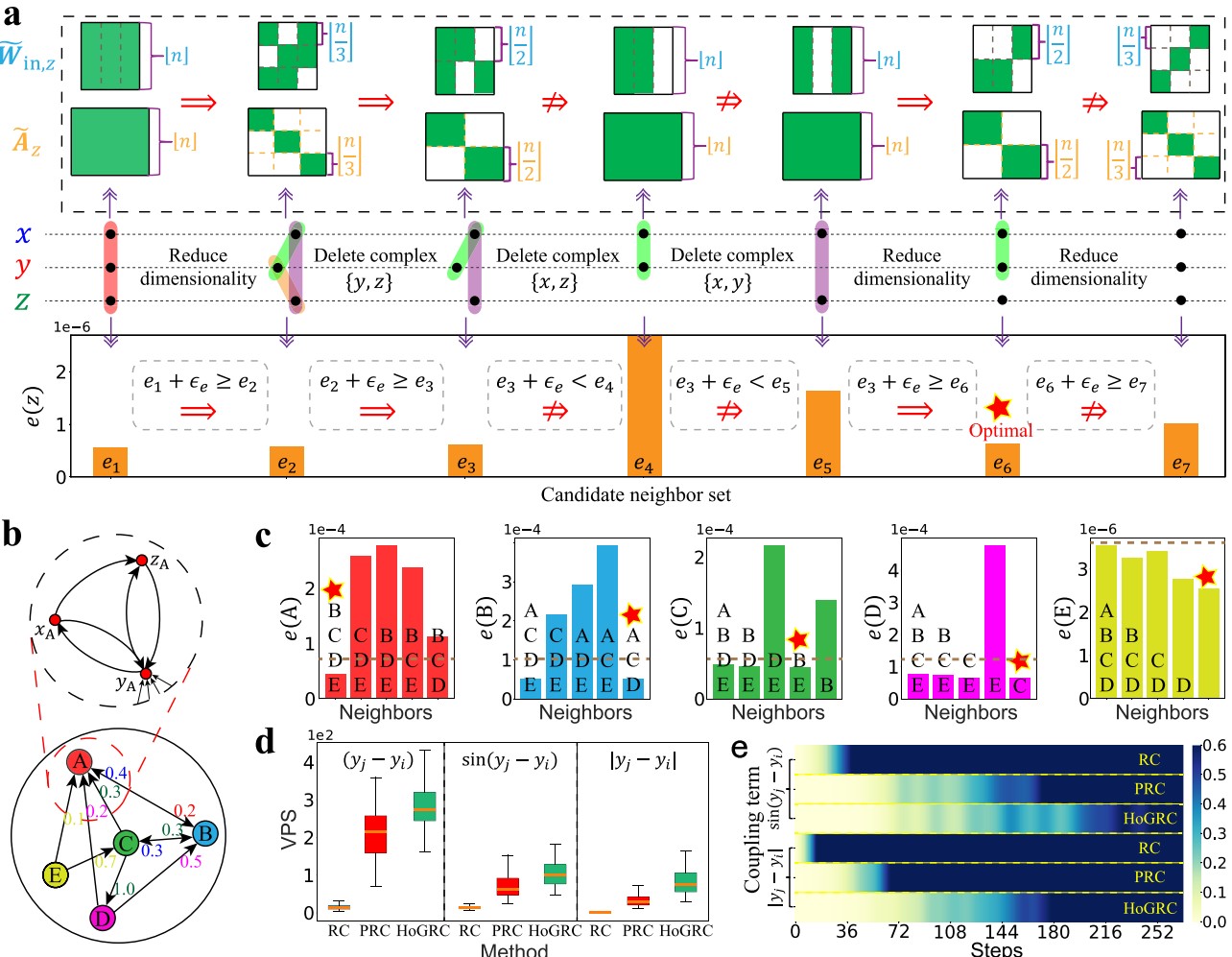

**Fig. 2 | Higher-order structure inference and dynamics prediction for the Lorenz63 system and the CL63 system. a** The successively iterative results on higher-order neighbors inference for node $z$ using Algorithm 1 in Table 1, where the red pentagram indicates the inferred higher-order neighbors $\mathscr{S}_z$. **b** The underlying coupling network of the CL63 system. **c** The inferred coupling neighbors for each subsystem of the CL63 system. **d**, **e** The average number of predictable steps and average prediction error of different methods for the nonlinear coupling cases, respectively. The orange line in the middle of the box represents the median, the upper and lower boundaries of the box represent the upper and lower quartiles, respectively. The boundaries of the upper and lower whiskers represent the maxima and minima, respectively. We set parameters as $\sigma = 10$, $\rho = 28$ and $\beta = 8/3$. Here, we use a time-step size $\Delta t = 0.02$ and a time-step number $T = 5000$.

where $m$ denotes the number of the subsystems, $h_i$ is the scale of the $i$-th subsystem, $\gamma$ represents the coupling strength, $w_{ij}$ is the coupling weight, and $g_{ij}$ denotes the coupling function. We consider a 15-dimensional CL63 system with 5 subsystems, and the structure and the coupling weights are depicted in Fig. 2b. We generate data with the coupling strength $\gamma = 0.5$ and the coupling function $g_{ij} = (y_j - y_i)$. Based on this data, we calculate the Lyapunov Exponents (LE's) of the system (see Supplementary Note 4.2), which suggests a higher-degree complexity emerging in the system, as more than half of the LEs are positive.

Our HoGRC framework considers complexes $\{y_i\}$ and $\{y_j\}$ as the higher-order neighbors of $x_i$ if subsystem $j$ has a coupling effect on $i$. Thus, by virtue of Definition 3, we are able to infer such a coupling relationship between any two subsystems. As depicted in Fig. 2c, we initially present the one-step prediction error for any subsystem $i$, considering all four other subsystems are treated as neighbors. Subsequently, we proceed to present the prediction errors when each neighboring subsystem is individually removed. The experimental results demonstrate that with the removal of subsystem $j$, the stronger the coupling effect of subsystem $j$ on $i$, the worse the prediction performance of subsystem $i$ is. This enables us to directly infer the

true interaction network among subsystems (marked by the red pentagrams).

For our second task, we perform multi-step predictions on the CL63 system using different methods. We randomly select 50 points from the testing data as starting points and use the predictable steps to quantify the prediction performances for the various methods. Figure 2d displays a boxplot of the predictable steps for various methods on 50 testing sets. The results clearly indicate that the HoGRC framework outperforms the other two methods, highlighting its superior ability in the extrapolation prediction. Furthermore, we extend our analysis by generalizing the linear coupling term $g_{ij} = (y_j - y_i)$ to two more nonlinear forms, namely $\sin(y_j - y_i)$ and $|y_j - y_i|$. Correspondingly, we include the complex $\{y_i, y_j\}$ in the higher-order neighbors of $x_i$. The heatmap of the prediction errors along with the time steps for various methods is illustrated in Fig. 2e. Combining with Fig. 2d, it becomes apparent that the HoGRC framework maintains its superiority in terms of prediction performance.

## Investigations on network dynamics
In recent years, network dynamical systems (NDS) have gained significant attention for their broad range of applications. As a special

form of system (1), NDS often exhibits a higher number of dimensions and more complex structural information. Therefore, our framework has become an efficient tool for NDS's structural inference and dynamic prediction. Generally, NDS's dynamics are modeled as:

$$\dot{\mathbf{x}}_i = F(\mathbf{x}_i) + \gamma \sum_{j=1}^{m} \omega_{ij} G(\mathbf{x}_i, \mathbf{x}_j), \tag{14}$$

where $\mathbf{x}_i = (x_i^1, \ldots, x_i^N)^\top$ denotes the $N$-D state of the $i$-th subsystem, $F$ represents the self-dynamics, $G$ represents the interaction dynamics, $\gamma$ is the coupling strength, $w_{ij}$ is the interaction weight of subsystem $j$ to $i$. Before presenting the results of our numerical investigations, we first make three remarks. (i) Since the HoGRC framework is a node-level based method, here we set the coupling network structure between any two subsystems as depicted in Fig. 2d. (ii) A very small coupling strength implies a weak coupling effect on the dynamics, while sufficiently strong coupling tends to increase predictability due to a high probability of synchronization occurrence (see Supplementary Note 4.6 for details). Therefore, in our investigations, we selected a moderate level of coupling strength to increase prediction difficulty. (iii) In addition to the RC and the PRC methods, we use two recently proposed powerful methods, namely the Neural Dynamics on Complex Network (NDCN)[15] and the Two-Phase Inference (TPI)[16], as the baseline methods for NDS predictions. The NDCN combines the graph neural networks with differential equations to learn and predict complex network dynamics, while the TPI automatically learns some basis functions to infer dynamic equations of complex system behavior for network dynamics prediction. Refer to Supplementary Note 5 for further details.

We first consider the coupled FitzHugh–Nagumo system (FHNS)[57] that describes the dynamical activities of a group of interacted neurons with

$$F(\mathbf{x}_i) = F(x_i^1, x_i^2) = \left(x_i^1 - (x_i^1)^3 - x_i^2, a + bx_i^1 + cx_i^2\right)^\top,$$
$$G(\mathbf{x}_i, \mathbf{x}_j) = G(x_i^1, x_j^1) = \frac{1}{k_i^{\mathrm{in}}}(x_i^1 - x_j^1), \tag{15}$$

in network dynamics (14). Here, we set $\gamma = 0.5$, $a = 0.28$, $b = 0.5$, $c = -0.04$, and $m = 5$ to generate experimental data. As shown in Fig. 3a, the trajectory predicted by our HoGRC framework closely matches the true trajectory of the FHNS system. In task (I), we begin by examining the inference of the coupling network among subsystems. Figure 3b displays the prediction errors for each subsystem under different coupling structures. The bar chart above includes multiple letters indicating the candidate neighbors of the corresponding subsystem. It is evident that the inferred coupling structures, illustrated with red pentagrams, align with our initial setting. Furthermore, in Supplementary Note 4.3, we provide the inference of higher-order neighbors for individual nodes within the subsystem as well, which further validates the effectiveness of our method. For task (II), we conduct the multi-step prediction experiments and compared our results to the baseline methods on 50 testing sets. The results, depicted in Fig. 3c, demonstrate that our method outperforms the other methods in terms of the extrapolation prediction performance.

We also investigate two other network dynamics, namely the coupled Rossler system (CRoS)[58] and the coupled simplified Hodgkin-Huxley system (CsH²S)[59]. The CRoS has the form

$$F(\mathbf{x}_i) = F(x_i^1, x_i^2, x_i^3) = \left(-h_i x_i^2 - x_i^3, h_i x_i^1 + a x_i^2, b + x_i^3(x_i^1 + c)\right)^\top,$$
$$G(\mathbf{x}_i, \mathbf{x}_j) = G(x_i^1, x_j^1) = x_j^1 - x_i^1, \tag{16}$$

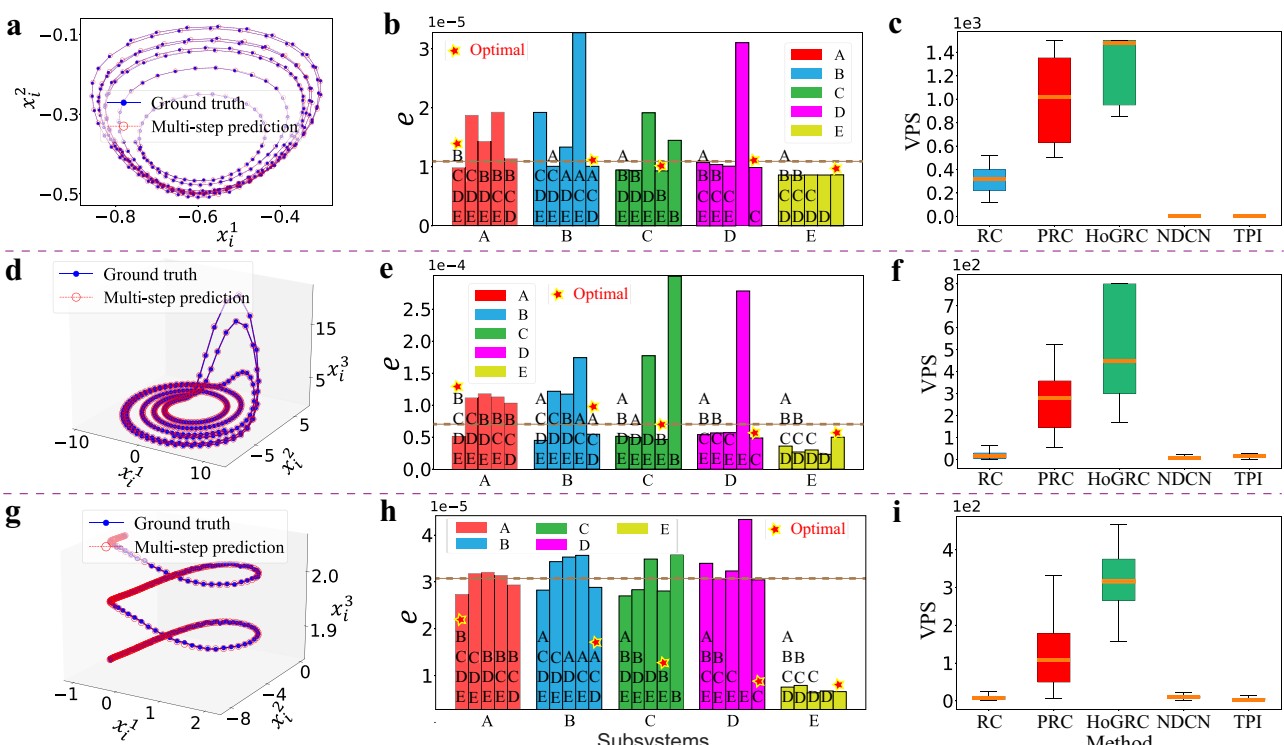

**Fig. 3 | Coupling network inference, system reconstruction, and dynamics prediction for network systems.** The corresponding results using the HoGRC framework and two other methods for the FHNS (**a**–**c**), CRoS (**d**–**f**), and CsH²S (**g**–**i**) network systems. The orange line in the middle of the box represents the median, the upper and lower boundaries of the box represent the upper and lower quartiles, respectively. The boundaries of the upper and lower whiskers represent the maxima and minima, respectively. The experimental data for these systems are generated by setting $T = 5000$ and using $\Delta t = 0.25$, 0.1, and 0.04, respectively.

in network dynamics (14), with $h_i$ representing the scale of the $i$-th subsystem, and with $a = 0.2$, $b = 0.2$, $c = -6$, $\gamma = 1$ and $m = 5$. The CsH$^2$S has the form

$$F(\mathbf{x}_i) = F(x_i^1, x_i^2, x_i^3)$$
$$= \left( x_i^2 - a(x_i^1)^3 + b(x_i^1)^2 - x_i^3 + I_{\text{ext}}, c - u(x_i^1)^2 - x_i^2, r[s(x_i^1 - x_0) - x_i^3] \right)^\top,$$

$$G(\mathbf{x}_i, \mathbf{x}_j) = G(x_i^1, x_j^1) = (V_{\text{syn}} - x_i^1) \cdot \mu(x_j^1), \mu(x) = \frac{1}{1 + e^{-\lambda(x - \Omega_{syn})}},$$
(17)

in network dynamics (14), with $a = 1$, $b = 3$, $c = 1$, $u = 5$, $s = 4$, $r = 0.005$, $x_0 = -1.6$, $\gamma = 0.1$, $V_{\text{syn}} = 2$, $\lambda = 10$, $\Omega = 1$, $I_{\text{ext}} = 3.24$, and $m = 5$. The investigation results, respectively, presented in Fig. 3d–f, g–i, suggest that our HoGRC framework possesses extraordinary capability in dynamics reconstructions and predictions using the inferred information of higher-order structures. It is noted that, in the examples above, the performances of the NDCN and the TPI are not satisfactory. This is because the NDCN is a network-level method that may not achieve good performance in complex nonlinear systems, and because the interaction function weights $w_{ij}$ in front of $G(\mathbf{x}_i, \mathbf{x}_j)$ are different, so the TPI method cannot learn the accurate basis function (refer to Supplementary Note 5 for the detailed illustration).

## Application to the UK power grid system

Finally, we apply the HoGRC framework to a real power system. We choose the UK power grid[60] as the network structure, which includes 120 units (10 generators and 110 consumers) and 165 undirected edges, as shown in Fig. 4a. To better describe the power grid dynamics, we consider a more general Kuramoto model with higher-order interactions[61], which can be represented as:

$$\dot{\theta}_i = \omega_i + \gamma_1 \sum_{j=1}^{N} A_{ij} \sin(\theta_j - \theta_i) + \gamma_2 \sum_{j=1}^{N} \sum_{k=1}^{N} B_{ijk} \sin(\theta_j + \theta_k - 2\theta_i), \quad (18)$$

where $\theta_i$ and $\omega_i$ denote the phase and natural frequency of the $i$th oscillator respectively, $\gamma_1$ and $\gamma_2$ are the coupling strengths, while pairwise and higher-order interactions are encoded in the adjacency matrix $A$ and adjacency tensor $B$. Under specific coupling settings, this kind of system exhibits extremely complex chaotic dynamics rather than synchronization.

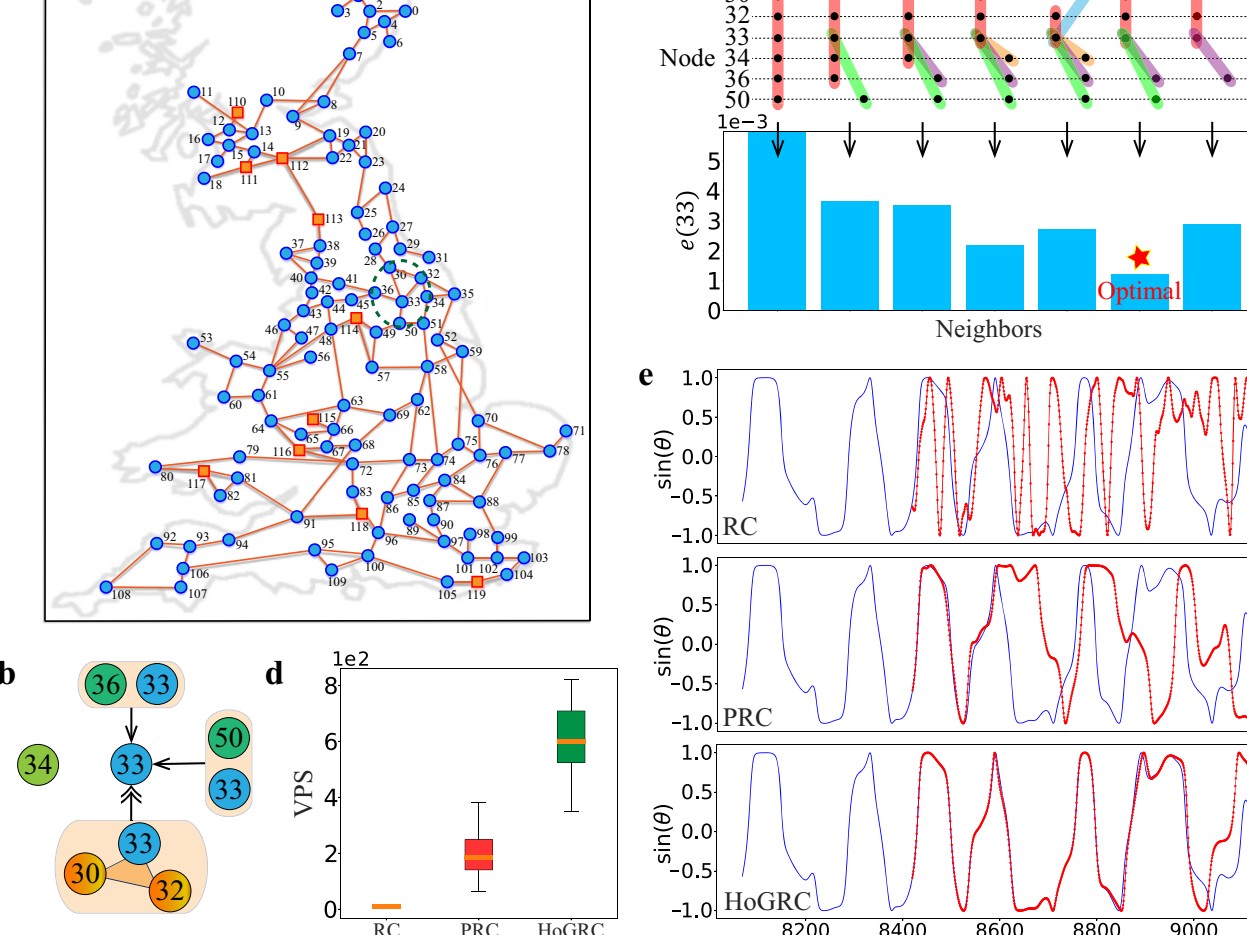

**Fig. 4 | Higher-order neighbors inference and dynamics prediction for the UK power grid system using the higher-order Kuramoto model. a** The UK power grid. **b** Local coupling structure of node 33. **c** Higher-order neighbors inference of node 33. **d** The average predictable steps of the entire system in the test set. The orange line in the middle of the box represents the median, the upper and lower boundaries of the box represent the upper and lower quartiles, respectively. The boundaries of the upper and lower whiskers represent the maxima and minima, respectively. **e** Extrapolation prediction of node 33 under different methods, with the true value shown in blue and the predicted value in red. We set $T = 10,000$, $\Delta t = 0.08$, $\gamma_1 = 0.4$, and $\gamma_2 = 0.4$.

Due to the special form of this model and the prediction challenges posed by higher-order terms, we need to apply a special treatment when using the HoGRC framework. We take the 2-D data $(\sin(\theta(t)), \cos(\theta(t)))$ as the input of the HoGRC framework at time $t$ and $\Delta\theta = (\theta(t+1) - \theta(t))/\Delta t$ as the output. Therefore, the predicted value in the next step is $\hat{\theta}(t+1) = \Delta\theta\Delta t + \theta(t)$. Thus, in multi-step prediction tasks, we can use the predicted value $(\sin(\hat{\theta}(t+1)), \cos(\hat{\theta}(t+1)))$ as the input for iterative prediction. For fairness, the RC and PRC methods also adopt the same treatment in the subsequent comparative tests.

To verify the advantages of our method, we consider the higher-order interactions which are constructed by identifying each distinct triangle from the UK power grid and generated data for the experiment, and Fig. 4b shows the local coupling network of node 33 (see Supplementary Note 4.4 for details of all higher-order interactions). Figure 4c shows the one-step prediction error for cases with different neighbors. We observe that the real higher-order neighbors correspond to the lowest prediction error. In the prediction task, our method outperforms the RC and PRC methods (see Fig. 4d, e), thanks to the structural complexity and high nonlinearity of the model, which make traditional methods prone to overfitting. Our method can learn the real dynamics of the system, leading to accurate predictions over a longer range.

### Different role of noise perturbation

Noise perturbation is a major factor that can affect the efficacy of any method in dealing with data. Hence, to demonstrate the robustness of our method against noise perturbations, we introduce noises of different intensities into the generated data.

In particular, we use Gaussian noise with zero mean and standard deviation $\sigma_n$ to introduce noise into the data. Empirically, due to the presence of noise, we increase the threshold $\epsilon_r$ to 0.03. Figure 5a shows the prediction performances for cases without and with added noise. With a certain level of noise intensity, such as $\sigma_n = 0.2$, our method is able to infer higher-order neighbors for both the Lorenz63 system and the CL63 system (refer to Supplementary Note 4.5 for specific details). Figure 5b–e shows the prediction performances when increasing noise intensity for the Lorenz63 and CL63 systems, while Fig. 5f shows the results for the hyperchaotic system (see Supplementary Note 2.2). Clearly, our method works robustly on data with noise intensity in a certain range.

To be candid, the excessive noise can adversely affect the accuracy of predictions across various examples. However, we interestingly find that in some cases, a moderate amount of noise can promote predictions, as shown in Fig. 5c–f. This type of noise can enhance the generalization ability of our method, especially when the HoGRC framework experiences overfitting issues even after sufficient training. If the structures or dynamics of the learned dynamical system are not too complex, the HoGRC framework after training can approximate the original dynamics with high fidelity. Nevertheless, noise generally has a negative effect.

### Influence of training set sizes and coupling network

Training set sizes and network structures are factors that significantly influence dynamic predictions. Typically, machine learning methods learn and predict unknown dynamics better with larger training set sizes or simpler network structures. Although all methods follow this general rule, our HoGRC method still has several advantages. To demonstrate this, we conduct the following numerical experiments.

On one hand, we use CRoS as an example to generate experimental data with different time lengths (other settings are the same as above). As shown in Fig. 6a, increasing the training data size initially improves prediction accuracy, which then levels off. Our method outperforms baseline methods even with a sufficient amount of training data, suggesting that our method can learn dynamics with fewer data points and more accurately capture real dynamical mechanisms. On the other hand, we investigate the impact of different network structures. We begin by considering regular networks with varying numbers of subsystems and generate experimental data using CRoS with a length of 5000 and $\Delta t = 0.1$. As shown in Fig. 6b, the network scale does affect the prediction accuracy in that, for a long-term prediction task, the prediction failure of one subsystem in the network can impact the prediction of the other subsystems via its neighbors. Compared to baseline methods, our method is less affected by network size and presents better predictability for large-scale systems. These advantages persist when considering the Erdös–Rényi (ER) networks[62] and the Barabasi-Albert (BA) networks[63] containing 30 subsystems, as demonstrated in Fig. 6c, e, f. Here, the average degrees of the regular, the ER, and the BA networks, respectively, are 2, 2.2, and 1.87. We randomly generate the coupling weights connecting every two subsystems in these networks.

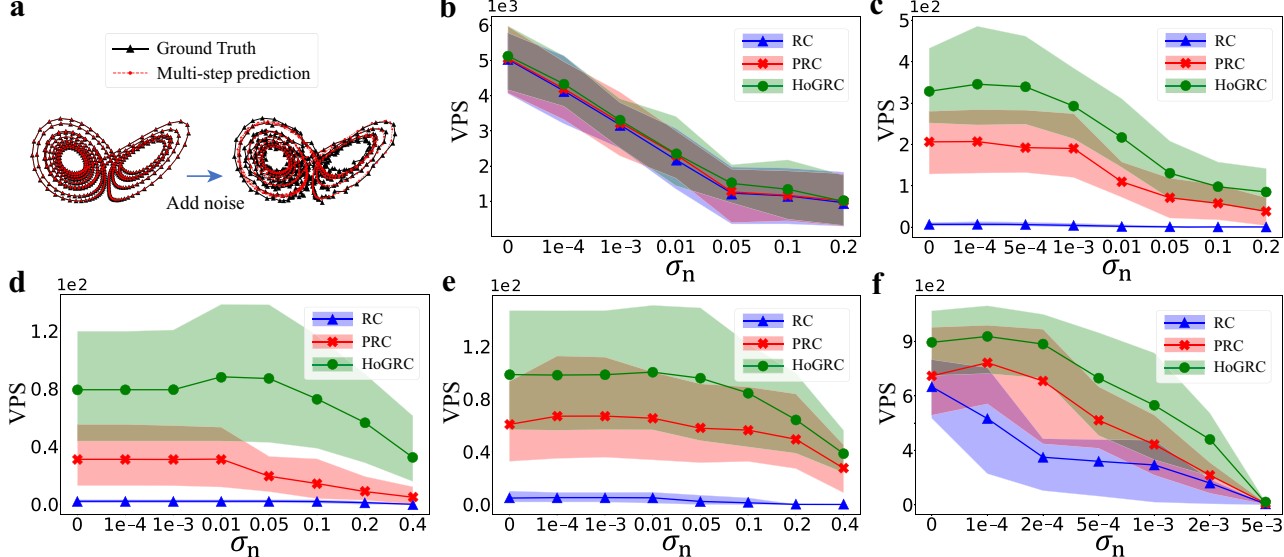

**Fig. 5 | Impact of noise on dynamics reconstruction and prediction using different methods. a** Dynamics reconstruction and prediction with and without noise for the Lorenz63 system. Prediction performances using different methods change with the noise intensity for the Lorenz63 system (**b**), for the CL63 system (**c–e**), where the corresponding coupling terms are selected, respectively, as $(y_j - y_i)$, $\sin(y_j - y_i)$, and $|y_j - y_i|$, and for the hyperchaotic system (**f**).

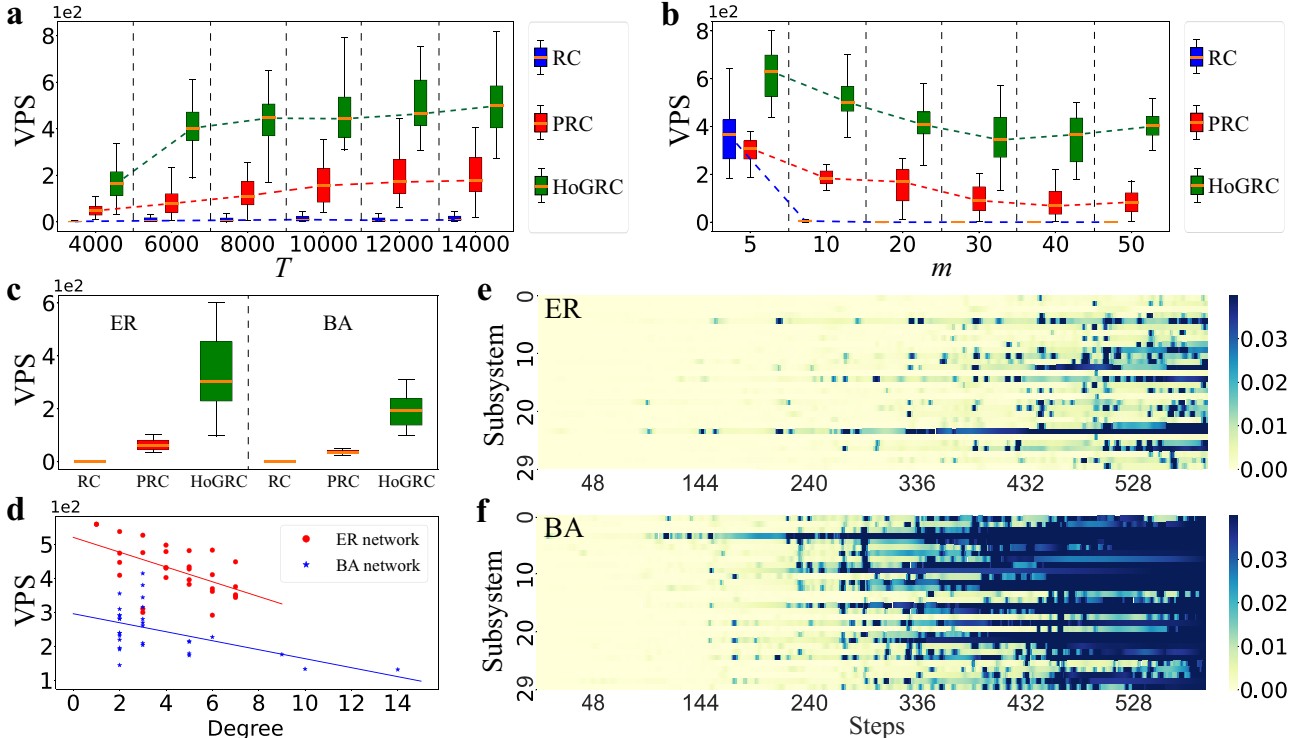

**Fig. 6 | Impact of training set sizes and system structures on dynamics prediction. a** The prediction performances with different training set sizes. **b** The prediction performances in the regular networks with varying numbers of subsystems. **c** The prediction performances in ER and BA networks with 30 subsystems. The orange line in the middle of the box represents the median, the upper and lower boundaries of the box represent the upper and lower quartiles, respectively.

The boundaries of the upper and lower whiskers represent the maxima and minima, respectively. **d** The predictable steps change with the degree of the subsystems, respectively, for ER and BA networks. **e, f** The prediction errors of the subsystems change with the time evolution using the HoGRC framework for the ER and the BA networks, respectively.

Additionally, from Fig. 6e, f, we interestingly find that, under the same average degrees, predicting the system using the BA network seems to be more difficult, while using the regular network makes prediction much easier. This finding is understandable since the degree distribution of the BA network follows a power law distribution, which creates more complex structures and more fruitful dynamics in the system. To further verify this finding, we use the degree of subsystems as an indicator to reveal the complexity of subsystems and depict different negative correlations between the number of predictable steps and the degree of each subsystem for different network settings, as shown in Fig. 6d.

### Direct and indirect causality

In our framework, the GC inference and the RC prediction are performed simultaneously and complement each other. Notably, the HoGRC framework does not require precise learning of the system structure through GC. Instead, our framework focuses on optimizing the coupling structures to further maximize the prediction accuracy. As a result, both direct and indirect causality can be inferred in the inference task. Despite this, our framework consistently and accurately infers the high-order structures in multiple experiments conducted in this study (see Supplementary Note 1.4 for specific reasons).

To further identify the direct and indirect causality, we can extend our HoGRC framework by combining it with the existing methods. In particular, we propose two strategies: (1) conditional Granger causality and (2) further causal identification. We provide the details of the above two strategies and experimental validation in Supplementary Note 1.4. The experimental results demonstrate the high flexibility and generality of our framework, enabling it to identify direct and indirect causality in conjunction with some existing techniques.

## Discussion

In this article, we have introduced a scalable HoGRC framework that is inspired by the classic idea of Granger causality and advances achieved in dynamics predictions using RC framework. Our proposed method facilitates accurate system reconstructions and long-term dynamics predictions by inferring higher-order structures at the node level. The method comprises of two inseparable tasks: high-order structure inference and multi-step dynamics prediction. To close this article, we provide the concluding remarks as follows.

First, in many complex chaotic systems, the system variables often lack mutual correlation. As a result, traditional methods may lead to false causality and negatively impact prediction accuracy. However, numerical experiments suggest that stronger coupling weights between dynamic causes make them more easily inferred. Nonetheless, weak coupling weights still have a non-negligible effect on prediction accuracy and require delicate methods such as the HoGRC framework. In addition, our framework possesses high flexibility and generality, allowing for further identification of direct and indirect causality by incorporating existing techniques.

Second, higher-order neighbors provide richer information than pairwise structures. This is because they not only eliminate non-causal signals but also significantly reduce the spurious interaction between causal signals. Compared to traditional methods, the HoGRC framework is better suited to accurately learning true dynamic mechanisms, thus avoiding overfitting during long-term predictions of dynamics. Additionally, the HoGRC's node-level prediction method allows for parallel implementation of inference and prediction tasks, making it ideal for large-scale system data. Particularly for complex coupling connections, where cause signals of nodes are intricate, the HoGRC framework shines, whereas traditional methods are prone to overfitting.

In terms of the future research topics, there are several areas of focus that warrant exploration. Firstly, it would be highly valuable to apply the newly proposed framework to a wider range of general dynamical systems with much more complex higher-order interaction structures. Additionally, there is a need to develop an efficient algorithm that can effectively eliminate the issue of indirect causality. Indeed, theoretical interpretations regarding this new framework would be much more meaningful, promoting us to further enhance the framework. Future extensions would combine our framework with the other advanced neural programming frameworks[64] and extend its application to more real-world complex systems. Overall, these future research directions will contribute to advancing our understanding of complex dynamical systems and improving the practicality, scalability, and robustness of the proposed framework.

## Methods

Here, we formulate the HoGRC framework by incorporating the higher-order structures that are possibly present in complex systems into the conventional RC method. To utilize higher-order neighbors precisely, we develop an algorithm inspired by the Granger causality. This renders the HoGRC framework applicable to both structure inference and dynamics prediction.

### From RC to higher-order RC

The traditional RC method comprises three parts, namely the input layer, hidden layer, and output layer. The $N$-D data $\mathbf{x}$ is embedded into a high-dimensional reservoir network at the input layer. Then, the $n$-D state sequence $\{\mathbf{r}(t)\}$ is obtained by specific rules within the reservoir as Eq. (2). Here, $\mathbf{W}_{\text{in}}$ and $\mathbf{A}$ are randomly generated and fixed, so we only need to train the parameter matrix $\mathbf{W}_{\text{out}}$ in the output layer. To better present our framework, we introduce an equivalent transformation here where we predict the difference instead of the next step value, given by Eq. (3). The ridge regression technique is generally used to obtain optimal $\mathbf{W}_{\text{out}}$ with the loss function as Eq. (4). However, the single RC method discussed above disregards the intrinsic correlation of the $N$-D input data and instead predicts the entire dynamics through training as a black box. This approach makes it challenging to unveil underlying dynamical structures in high-dimensional complex systems.

To address this limitation, a parallel local strategy PRC based on entropy causality was later proposed[39,40]. In the PRC approach, a directed edge from node $v$ to $u$ is connected and deemed a dynamic causal link if the dynamic equation of node $u$ contains $v$. However, this approach only incorporates the pairwise structures at the most elementary level for characterizing complex systems. Instead, we integrate $u$ and all its different order of neighbors as inputs into the input layer, thereby enhancing the prediction of node $u$.

In order to enhance the accuracy of reconstructing and predicting complex dynamics from the observational data, it is crucial to integrate the higher-order structures into our model. In light of this, we propose the HoGRC framework that integrates these structures. Specifically, for any node $u$ within the system, analogous to Eq. (2), the hidden dynamic at the node-level in the HoGRC framework is given by (8), where the key of the structure input lies in encoding the higher-order neighbors into the input and the adjacency matrices of the hidden dynamics, denoted as $\tilde{\mathbf{W}}_{\text{in},u}$ and $\tilde{\mathbf{A}}_u$ (see settings in (9)). In addition, the higher sparsity in $\tilde{\mathbf{W}}_{\text{in},u}$ and $\tilde{\mathbf{A}}_u$ in the HoGRC framework eases the learning task and minimizes overfitting. We provide theoretical explanations through the following proposition, assuming that different RC methods share the same hyperparameters (see Supplementary Note 1.1 for its proof).

**Proposition 1.** Assuming that the input matrix and the adjacency matrix in different RC models are generated by the same random

method. Then,

$$\mathscr{H}_{\text{HoGRC}} \subseteq \mathscr{H}_{\text{RC}}, \tag{19}$$

where $\mathscr{H}_{\text{RC}}$ and $\mathscr{H}_{\text{HoGRC}}$ denote the sets of the hidden dynamical systems modeled by Eq. (2) in RC and by Eq. (8) in HoGRC, respectively. Furthermore, if the dataset has an upper bound, denoted by $B$, on its potential distribution $\mathscr{D}$, i.e.,

$$\max_{\mathbf{x} \sim \mathscr{D}} \| \mathbf{x} \|_{\infty} \leq B, \tag{20}$$

where $\mathbf{x}$ is the $N$-D data. Then, the HoGRC framework has a smaller upper bound of the generalization error, that is,

$$GE_{\text{u}}(h_{\text{HoGRC}}) \leq GE_{\text{u}}(h_{\text{RC}}), \tag{21}$$

where $h_{\text{HoGRC}} \in \mathscr{H}_{\text{HoGRC}}$, $h_{RC} \in \mathscr{H}_{RC}$, and $GE_u(h)$ denotes the upper bound on the generalization error when reconstructing the original dimension $u$ using the hidden dynamical system $h$.

### Structures inference and dynamics prediction

As mentioned earlier, our framework aims to leverage information from higher-order neighbors for prediction. However, in practice, the structure information is often unknown a priori, necessitating the inference of higher-order causal links connecting nodes before making predictions. Consequently, the HoGRC possesses a two-folded mission: Higher-order neighbors inference and dynamics prediction using the inferred higher-order structures.

**Task (I):** *Inferring higher-order neighbors.* Since higher-order interactions are inherently complex and nonlinear, the classic Granger causality method cannot be directly applied but brings us some inspiration. To this end, we consider the case where node $u \in V$ awaits prediction, so we have

$$u(t) = q\big(\{\mathbf{c}_1, \mathbf{c}_2, \dots, \mathbf{c}_K\}(\leq t)\big) + \mathbf{e}_t, \tag{22}$$

where $q$ is the prediction function represented by the HoGRC method, $\mathscr{C} = \{\mathbf{c}_1, \dots, \mathbf{c}_K\}$ is the candidate complex set containing higher-order neighbors of node $u$, and $q(\mathscr{C}(\leq t))$ represents the one-step prediction result obtained by inputting higher-order structure $\mathscr{C}$ and the observed data $\mathbf{x}$ before time $t$. Then we can define the mean prediction error as

$$e_{\{\mathbf{c}_1, \dots, \mathbf{c}_K\}}(u) = \frac{1}{T} \sum_t |q\big(\{\mathbf{c}_1, \mathbf{c}_2, \dots, \mathbf{c}_K\}(\leq t)\big) - u(t + \Delta t)|, \tag{23}$$

where $T$ denotes the length of the data. In this context, excluding the Granger causality from $\mathbf{c}_k$ to $u$ implies that the function $q$ does not depend on $\mathbf{c}_k$. We formally define this concept as follows.

**Definition 3.** Assume that all the higher-order causal links for node $u$ are included in the candidate set $\{\mathbf{c}_1, \dots, \mathbf{c}_K\}$. Also, assume that $\mathbf{c}_k$ is not a subcomplex of any other candidate simplicial complex and further that the inequality

$$e_{\{\mathbf{c}_1, \dots, \mathbf{c}_K\}}(u) + \epsilon_{\text{e}} \geq e_{\{\mathbf{c}_1, \dots, \mathbf{c}_{k-1}, \mathbf{c}_{k+1}, \dots, \mathbf{c}_K\}}(u) \tag{24}$$

is satisfied. Then, the simplicial complex $\mathbf{c}_k$ is not the causal factor in Granger's sense for node $u$, where $\epsilon_{\text{e}}$ is a threshold taking positive value. That is, the complex $\mathbf{c}_k$ is not a higher-order neighbor of node $u$.

In truth, other metrics may also be used to evaluate prediction performance. We propose a greedy strategy that searches for the exact higher-order neighbors and filters candidate complexes in order of decreasing dimension and importance. The algorithmic process is briefly outlined in Algorithm 1 of Table 1, with additional details about

the algorithm and the selection of the threshold $\epsilon_e$ provided in Supplementary Note 1.2.

**Task (II)**: *Predicting dynamics using the HoGRC framework*. Using the inferred higher-order interactions, we provide data for each node and its higher-order neighbors to the HoGRC, which then predicts subsequent values over time. By continually adding the most recent forecasted values to the input data, we can make multistep-ahead predictions.

## Data availability

All the datasets generated in this study have been deposited in the Github database under the accession code in "dataset" folder in GitHub repository: https://github.com/CsnowyLstar/HoGRC[https://doi.org/10.5281/zenodo.10685733][65].

## Code availability

The code used in this study is freely available in the public GitHub repository: https://github.com/CsnowyLstar/HoGRC[https://doi.org/10.5281/zenodo.10685733][65].

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

## Acknowledgements

Q.Z. is supported by the China Postdoctoral Science Foundation (No. 2022M720817), by the Shanghai Postdoctoral Excellence Program (No. 2021091), and by the STCSM (Nos. 21511100200, 22ZR1407300, and 22dz1200502). W.L. is supported by the National Natural Science Foundation of China (No. 11925103) and by the STCSM (Nos. 22JC1402500, 22JC1401402, and 2021SHZDZX0103). H.M. is supported by the National Natural Science Foundation of China (No. 12171350). The computational work presented in this article is supported by the CFFF platform of Fudan University.

## Author contributions

W. Lin conceived the idea. C. L. Zhao and Q. X. Zhu designed the research and helped perform the analysis with constructive discussions. X. Li performed the experiments and wrote the initial draft of the manuscript. X. Zhang and B. L. Zhao collected the data and carried out additional analyses. X. J. Duan, H. F. Ma, and J. Sun contributed to refining the ideas. All authors contributed to writing the manuscript.

## Competing interests

The authors declare no competing interests.
