## [Peer Review File · Nature Communications]

Higher-order Granger reservoir computing: Simultaneously achieving scalable complex structures inference and accurate dynamics predictionReviewers' comments:

Reviewer #1 (Remarks to the Author):

This paper is very difficult to read, there are many concepts which are not clearly defined. Moreover the comparison with other works could be clearer.

The authors need to explain clearly what are the differences with the following paper, which is from my point of view, far clearer and easier to understand than this one.

<https://www.nature.com/articles/s42256-023-00668-8>

Moreover, because the code is available for this paper, you should compare your approach with this one.

So I expect the authors to propose a better version of their paper in which they clearly explain the advantages of their approach compared to the other approaches. Moreover, they need to give a reproducible and understandable code that allows researchers to understand their approach.

Fig 1 is not understandable. HoGRC is not defined. So what should the reader understand?

Fig 2 has the same problem. There are so many things without any explanation...

I am not able to understand anything in all the figures.

Some typos (there are probably many other ones)

page 5

Additionally, we adopte the root mean square error (RMSE) as a metric to quantitatively assess prediction error

page 12

To be candid, noise with sufficiently large intensity can negatively impact prediction accuracy in all examples.

page 15

To more precisely reconstruct complex dynamics, it is imperative to incorporate additional higher-order structures.

Reviewer #2 (Remarks to the Author):

This work suggests to use a granger-causality concept to develop a pattern of interactions in a dynamical systems to better decide an underlying model. It uses reservoir computing as the basic engine behind prediction that underlies granger-causality, to decide variation of

forecasting with and without a certain factor in place. The call this HoRC, although I wonder why the phrase high order is merited. I have several worried, some related to technical details and several related to existing literature, some of which even includes one of the authors. I do not feel this paper is ready to publish at this time.

Reservoir computing is an excellent way to efficiently make data-driven forecasts of a dynamical system. However, isn't that exactly what is done in the paper [48] (they cite it, but not in a reasonable or substantive way. They cite it as a source for the famous Lorenz equations, but in fact, in [48], they perform exactly the same step of using dual RC forecasts to decide Granger-causality).

Furthermore, what if the interaction between elements were not so simple as Lorenz, which is truly a simple system, but one where it is necessary to decide between direct or indirect influences of the sort $X \rightarrow Y \rightarrow Z$? It does not seem from my reading that this approach can decide if $X \rightarrow Z$ directly or only through Y . This is a classic and perhaps subtle issue, but it surprises me that the authors do not mention it since one of the co-authors is the lead author on several papers that is premised on exactly this issue, Sun, Jie, Dane Taylor, and Erik M. Bollt. "Causal network inference by optimal causation entropy." *SIAM Journal on Applied Dynamical Systems* 14.1 (2015): 73-106, But by entropy means. And several author papers by the co-author. The basic issue does not go away whether it is by entropy methods or by reservoir prediction. This is a fundamental issue.

Also, the idea of using causation inference as a precursor associated with equation fitting exists in the literature, and here too, one of the co-authors is again a co-author on that work, so it is surprising that this current work is built as if this is a new idea. This is again substantial and not just something a new citation added will fix, since the central premise of this current paper already exists in the co-authors prior work. AlMomani, Abd AlRahman R., Jie Sun, and Erik Bollt. "How entropic regression beats the outliers problem in nonlinear system identification." *Chaos: An Interdisciplinary Journal of Nonlinear Science* 30.1 (2020), and again several author papers by the co-author, and others where the main premise that causation inference can be used to decide sparsity structure associated with modelling for efficient modelling. Rather than cite a half dozen papers for the last paragraph, and this one, since the author of this manuscript is on them, I will remind them, not just to cite, but to address the issue of deciding what is distinct other than RC is used, and as such how is it substantially different from [48].

Finally, and also importantly, whether fitting the equations directly without sparsity structure, or fitting by emphasizing sparsity structure as is done with SindY, or by the methods just described, its not a matter of this method got the right answer, especially for the simple problems illustrated here. It's a matter of efficiency, likely in terms of computational complexity. How well does this method do, and compare to the others, in terms of fitting quality in some measure of quality, with respect to growing data set size.

For these reasons I am not recommending publication.

Reviewer #3 (Remarks to the Author):

The authors present an original technique to analyse higher order complex networks by a promising combination of Granger causality with an extended kind of reservoir computing. They demonstrate the potential of their new method for various model systems of low and

intermediate dimension as well as for the UK power grid. Compared to other methods theirs outperforms them clearly in the prediction and numbers of iteration steps.

This is a very interesting and clearly written paper with much novelty. Therefore I recommend it for publication after a minor revision.

The authors should consider the following points:

- the coupling strengths used are rather large (0.4 ... 0.5). What about smaller ones which are often typical in real systems? Is there a minimum coupling strength which they can retrieve?
- I do not understand how many layers they have used (only n is given as the number of nodes in the reservoir). How many parameters to estimate for their algorithm?
- The model for power grids does NOT include losses which are very present in real ones (e.g. Nat. Comms. 11, s41467-020-14417-7 (2020)). This should be included in the model as well.
- In Fig. S6 the number of positive LEs should be given explicitly because it is almost impossible to estimate from the Fig.

Reply to the Reviewer’s reports

To Reviewer #1’s Report:

We sincerely appreciate the efforts made by the reviewers in thoroughly reviewing our work. Accordingly, we have taken your valuable suggestions into careful consideration and have meticulously addressed each concern point by point. This has significantly contributed to enhancing the overall readability of the revised manuscript. Furthermore, we have supplemented the additional experiments and the source code to further clarify the contributions of our work. Our revised manuscript has been uploaded, with the changes clearly highlighted in red for ease of reference.

Comment 1

This paper is very difficult to read, there are many concepts which are not clearly defined. Moreover the comparison with other works could be clearer.

Reply. Many thanks for your valuable comments. We recognize that the original version lacked a bit of clarity in terms of content organization, method exposition, and crucial experimental descriptions which made it difficult for readers to fully understand the innovations and contributions of our work. As a result, we have undertaken a thorough revision of the manuscript to address these concerns.

Although all newly introduced concepts in our manuscript have definitions in the **Methods** section, we acknowledge that the description of the proposed HoGRC framework in the main text was too rigid, leading to decreased readability. To address this issue, we thoroughly refined the presentation of the HoGRC framework in the main text, particularly in the final paragraph of the **Introduction**, **Section 2.1**, and **Section 2.2**. Additionally, we have a more comprehensive, clear, and coherent exposition of the framework in conjunction with the **Methods** section.

When choosing baseline methods, we conducted a comparative analysis that included not only the classical reservoir computing (RC) but also the state-of-the-art (SOTA) extensions

published in top conferences or journals in recent years (references [12], [13], [33], and [34] in the manuscript). To ensure a clearer presentation of the experimental results, we have enhanced the descriptions of these findings in the main text, and we have provided further details of the baseline methods in Section 5 of the Supplementary Material.

Comment 2

The authors need to explain clearly what are the differences with the following paper, which is from my point of view, far clearer and easier to understand than this one. <https://www.nature.com/articles/s42256-023-00668-8> Moreover, because the code is available for this paper, you should compare your approach with this one. So I expect the authors to propose a better version of their paper in which they clearly explain the advantages of their approach compared to the other approaches. Moreover, they need to give a reproducible and understandable code that allows researchers to understand their approach.

Reply. Thank you for your valuable advice. We have carefully read the paper you mentioned, which was recently published in *Nature Machine Intelligence* (**just three weeks prior to our work submission**), along with its associated code. It is important to note that while our work and theirs are both based on the reservoir computing (RC), the goals and focuses are quite different.

In particular, the main objective of their work is to propose a new computing paradigm based on RC, to decompile, code, and compile analogue computations. Technically, this approach involves a linear approximation of RC dynamics and employs low-order Taylor expansion to establish a framework for compiling and decompiling between the *known* system and the RC dynamics. More precisely, they represented the known dynamics equations as the **source code matrix** O , obtained the **programming matrix** R from the approximate RC dynamics, and subsequently obtained the **machine code** $W = \operatorname{argmin}_W \|WR - O\|$. Importantly, their neural machine code and programming framework **does not require any example data or sampling of state space**. Additionally, in their paper, it was also pointed out that one of the main limitations of their framework is the linear approximation of the RC dynamics. This approximation may not adequately capture nonlinear behaviors,

thus limiting the framework’s expressive capacity and predictive ability.

However, our work, from a different point of view, mainly focuses on two hot topics, namely structure inference and time series forecasting. Therefore, our work differs significantly from theirs in terms of task objectives. Specifically, we aim to integrate the spatial information, especially the higher-order structural information, into RC to significantly facilitate the prediction of unknown nonlinear dynamic systems. In turn, such significant enhancement inspires us to leverage the concept of Granger Causality to infer the higher-order structure of these systems. To the best of our knowledge, this is the first work to systematically analyze the impact of spatial information, the higher-order structures, on RC, and we provide a spatiotemporal reservoir framework that enables more accurate modeling of nonlinear dynamical systems from the observational data.

In conclusion, it appears that their framework may not be suitable for our specific tasks. Therefore, we add this reference to our revised paper to explore future directions, such as combining our framework with their neural machine code and programming framework to achieve scalable and effective decompile, code and compile analogue computations.

To highlight the superiority of our approach over other baselines, we have made substantial improvements to the relevant sections, such as the **Introduction**, **Results**, and **Methods** sections (refer to the revised paper for specifics). Furthermore, we have provided easily reproducible and comprehensible code for all experiments conducted in the study (please see the website <https://github.com/CsnowyLstar/HoGRC> for further information). We believe that the revised paper is a clearer and more readable version, enabling a better understanding of our work and its contributions.

Comment 3

Fig 1 is not understandable. HoGRC is not defined. So what should the reader understand?

Reply. Many thanks for your valuable comment. Accordingly, we have made significant improvements to enhance the clarity of Fig. 1 of the main text, and we have meticulously

Figure R1: **Schematic diagrams for illustrating the differences among the RC, PRC, and HoGRC frameworks.** **a, b** The execution processes for traditional RC and PRC methods, respectively. **c** The higher-order neighbors for the Lorenz63 system. **d** The connecting matrices \tilde{W}_{in} and \tilde{A} for the Lorenz63 system. **e** The higher-order Granger RC (HoGRC) combines Granger causality and higher-order structures in a mutually reinforcing cycle.

revised and redrawn this figure, ensuring that the illustrations are now much clearer and more easily understandable. Additionally, we have provided a more comprehensive definition of HoGRC. We believe that the revised version of the paper adequately addresses your concerns.

Specifically, a better understanding of the HoGRC framework can be obtained from the following two subsections.

1. A novel paradigm of RC with spatial information.

Figure R1(a) provides a concise representation of the learning process in traditional RC. To improve the performance of RC in higher-dimensional systems, recent studies (references [34],[35] in the revised paper) employ a node-level prediction technique depicted in Fig. R1(b).

Specifically, it involves feeding both the target node u and its neighbor nodes into RC, and finally decode the hidden state \mathbf{r} to the target node space x_u using the output matrix \mathbf{W}_{out} , resulting in more accurate dynamics predictions. This approach, known as PRC, allows for parallel predictions of the states of multiple nodes.

In this work, we present a novel computing paradigm for RC that incorporates spatial information, particularly focusing on the higher-order interactions. Specifically, we introduce the concept of higher-order neighbors using the simplicial complex to describe the higher-order structures within the system (see Methods section for details). To illustrate the concept, we provide a clear example in Fig. R1(c) that demonstrates the higher-order neighbors of all nodes and their corresponding binary representations. In general cases, we denote the set of higher-order neighbors of node u as $\mathcal{O}_u = \{o_1, o_2, \dots, o_{D_u}\}$, where o_i is a simplicial complex, corresponding to the i -th higher-order neighbor, and D_u is the higher-order degree of node u . To incorporate the spatial information into the RC, we encode it into the input and adjacency matrices, denoted as $\tilde{\mathbf{W}}_{\text{in}}$ and $\tilde{\mathbf{A}}$, respectively, as follows:

$$\begin{aligned}\tilde{\mathbf{W}}_{\text{in}} &= [\psi(o_1), \psi(o_2), \dots, \psi(o_{D_u})]^\top, \\ \tilde{\mathbf{A}} &= \text{diag}\{\varphi(o_1), \varphi(o_2), \dots, \varphi(o_{D_u})\},\end{aligned}\tag{1}$$

where $\varphi(o_i)$ represents a random sparse matrix of dimension $\lfloor n/D_u \rfloor \times \lfloor n/D_u \rfloor$ (where $\lfloor \cdot \rfloor$ is the floor function), and $\psi(o_i)$ represents a random sparse matrix of dimension $N \times \lfloor n/D_u \rfloor$. Only the signals of nodes contained in o_i serve as inputs, as shown in Fig. R1(d) for a particular case of the Lorenz63 system. By incorporating higher-order spatial information in this manner, the proposed higher-order RC model can more accurately capture the dynamics of complex systems.

2. Structure inference and dynamics prediction in HoGRC.

The specific form as well as the higher-order structures of a system are often unknown *a priori*. In light of this, we incorporate the concept of Granger causality (GC) into the higher-order RC to infer the underlying higher-order structures. Subsequently, the inferred structures are integrated into the higher-order RC model to further improve its prediction

performance. This iterative procedure is repeated until the model achieves optimal prediction accuracy. Consequently, we refer to this integrated model as the higher-order Granger RC (HoGRC) framework, as shown in Fig. R1(e).

In particular, we consider the set of candidate complexes for node u , denoted as $\mathcal{C}_u = \{c_1, c_2, \dots\}$, where c_i represents the i -th candidate complex for node u . It is important to note that $\mathcal{O} \subseteq \mathcal{C}$. To determine the non-Granger causality, we adopt the concept of GC and introduce Definition 3 of the main text. In essence, if the removal of node c_i does not significantly impact the prediction accuracy of node u , then c_i is not regarded as a higher-order neighbor of node u . Based on this concept, we develop an efficient greedy strategy, outlined in Algorithm 1 of the main text, to effectively explore and identify the set of higher-order neighbors for any given node u in an iterative manner. In our experiments, this algorithm exhibits remarkable performance, successfully inferring the higher-order structures of the systems.

Notably, in our HoGRC framework, the processes of GC inference and RC prediction are complementary and mutually reinforcing. Specifically, the structure discovered by GC can significantly enhance the predictive power of RC, and conversely, the novel RC with spatial information can aid GC in effectively discovering the structure. As a result, our framework can simultaneously achieve two tasks: (I) structures inference and (II) dynamics prediction. Similar to the PRC, due to the node-level nature of our framework, these tasks can be executed in parallel across all nodes. For more detailed information about HoGRC, please refer to Sections 2.1, 2.2, and Methods of the revised paper.

Comment 4

Fig 2 has the same problem. There are so many things without any explanation... I am not able to understand anything in all the figures.

Reply. Thank you for your valuable comment. We have taken your feedback into consideration and made revisions to enhance the readability of the manuscript. In particular, we have made improvements to the demonstrations presented in Figure 2 of the main text,

Figure R2: **Higher-order structure inference and dynamics prediction for the Lorenz63 system and the CL63 system.** **a** Higher-order neighbors inference results of the node z based on Algorithm 1, with the red pentagram indicating inferred higher-order neighbors. **b** The underlying coupling network of the CL63 system. **c** The inferred coupling neighbors for each subsystem of the CL63 system. **d**, **e** The average number of predictable steps and average prediction error of different methods for the nonlinear coupling cases, respectively.

ensuring that they are clearer and easier to understand. Additionally, we have tried our best to enhance the descriptions of both our methods and experiments. We hope that these improvements address your concerns.

Specifically, as illustrated in Fig. R2(a), we utilize the decimal numbers of binary encoding presented in Fig. R1(c) to represent the higher-order neighbors. For example, the decimal value “7” corresponds to the binary encoding “111”, which represents the highest-order neighbor composed of $\{x, y, z\}$. Subsequently, one can employ Algorithm 1 of the main text to infer the higher-order neighbors of all nodes in the Lorenz63 system. During the inference process, the one-step prediction errors of each node using different candidate complexes are shown in the bar graphs of Fig. R2(a). It is apparent that by appropriately selecting a threshold $\epsilon_e = 1.0 \times 10^{-7}$, our method can accurately infer the higher-order neighbors of each

node (highlighted by the red pentagram). In addition, the inferred results in the x and y dimensions can be found in Section 4.1 of the supplementary materials.

Then we consider a 15-dimensional CL63 system with 5 subsystems, and the structure and the coupling weights are depicted in Fig. R2(b). Our HoGRC framework considers complexes $\{y_i\}$ and $\{y_j\}$ as the higher-order neighbors of x_i if subsystem j has a coupling effect on i . Thus, we can infer the coupling relationship between any two subsystems in a similar manner. As depicted in Fig. R2(c), we initially present the one-step prediction error for any subsystem i , considering all four other subsystems are treated as neighbors. Subsequently, we proceed to present the prediction errors when each neighboring subsystem is individually removed. The experimental results demonstrate that with the removal of subsystem j , the stronger the coupling effect of subsystem j on i , the worse the prediction performance of subsystem i is. This enables us to directly infer the true interaction network among subsystems (marked by the red pentagrams).

For the prediction task, we perform multi-step predictions on the CL63 system using different methods. We randomly select 50 points from the testing data as starting points and use the predictable steps to quantify the prediction performances for the various methods. Figure R2(d) displays a box plot of the predictable steps for various methods on 50 testing sets, where the orange horizontal line represents the median. The results clearly indicate that the HoGRC framework outperforms the other two methods, highlighting its superior ability in the extrapolation prediction. Furthermore, we extend our analysis by generalizing the linear coupling term $g_{ij} = (y_j - y_i)$ to two more nonlinear forms, namely $\sin(y_j - y_i)$ and $|y_j - y_i|$. Correspondingly, we include the complex $\{y_i, y_j\}$ in the higher-order neighbors of x_i . The heatmap of the prediction errors along with the time steps for various methods is illustrated in Fig. R2(e). Combining with Fig. R2(d), it becomes apparent that the HoGRC framework maintains its superiority in terms of prediction performance. For more detailed information, please refer to Section 2.3 of the revised paper.

Comment 5

Some typos (there are probably many other ones)

page 5: Additionally, we adopte the root mean square error (RMSE) as a metric to quantitatively assess prediction error.

page 12: To be candid, noise with sufficiently large intensity can negatively impact prediction accuracy in all examples.

page 15: To more precisely reconstruct complex dynamics, it is imperative to incorporate additional higher-order structures.

Reply. Many thanks for your careful reading and helpful advice. We have revised the typos that you pointed out and conducted a thorough review of the entire manuscript to ensure the accuracy and rigor.

To Reviewer #2's Report:

We sincerely appreciate the time and effort you devoted to reviewing our work. Your comprehensive, valuable, and insightful comments helped us significantly improve the quality of this work. We acknowledge that certain aspects of our writing may have led to misunderstandings, and we apologize for any confusion caused. Accordingly, we try our best to address all of your specific concerns thoroughly and carefully. To facilitate the revision process, we upload a revised version of our manuscript with modifications highlighted in red.

Comment 1

This work suggests to use a granger-causality concept to develop a pattern of interactions in a dynamical systems to better decide an underlying model. It uses reservoir computing as the basic engine behind prediction that underlies granger-causality, to decide variation of forecasting with and without a certain factor in place. The call this HoGRC, although I wonder why the phrase high order is merited. I have several worried, some related to technical details and several related to existing literature, some of which even includes one of the authors. I do not feel this paper is ready to publish at this time.

Reply. Many thanks for your valuable comment. In this work, we mainly focus on two key tasks: structure inference and time series prediction. In fact, compared to the task of structure inference, we place more emphasis on the system reconstruction and extrapolation prediction using the newly proposed framework. Specifically, the core contribution of our work involves incorporating the spatial information, particularly the higher-order structural information, into the reservoir computing (RC) to significantly facilitate the prediction of nonlinear dynamical systems. In turn, such significant enhancement inspires us to leverage the idea of Granger causality to infer the higher-order structure of the underlying systems. To the best of our knowledge, this is the first article to systematically analyze the impact of higher-order structure information on enhancing the performance of RC. Additionally, we provide a spatiotemporal reservoir framework that enables more accurate modeling of dynamical systems using observational data.

In our HoGRC framework, Granger causality (GC) inference and RC prediction are com-

plementary and mutually reinforcing. Specifically, the structure discovered by GC can significantly enhance the predictive capability of RC. Conversely, our higher-order RC framework facilitates GC in effectively discovering the underlying structure. We subsequently iterate through this procedure until optimal prediction performance of HoGRC is attained. In fact, in this process, RC does not require a precise learning of the system structure through GC. This is because that our framework mainly focuses on optimizing the coupling higher-order structures to maximize predictive performance. Therefore, in the inference task, both direct causality and indirect causality may be inferred. Despite this, our method can accurately infer high-order structures in multiple experiments throughout the manuscript. Moreover, we extend our framework via integrating the existing methods and our inference strategies to further eliminate the indirect causality. For more details on the reasons for good inference performance and the extended method for handling the indirect causality, please refer to the **Reply to Comment 3**.

Moreover, it should be noted that our framework differs from previous state-of-the-art techniques (references [12], [13], [33], and [34] in the manuscript). Unlike these approaches, our framework is the first to integrate higher-order structures into RC to facilitate the prediction capabilities in dynamical systems. In the **Methods** section, we introduce the novel concept of higher-order neighbors using simplicial complexes, which provides a convenient way of describing the higher-order structure of a system. This higher-order structure, in contrast to the conventional pairwise structure, provides a more comprehensive and precise understanding of the intrinsic mechanisms governing dynamical systems, ultimately improving the predictive accuracy via the newly proposed framework.

In conclusion, we have made significant improvements to the **Introduction**, **Results**, and **Methods** sections in order to better illustrate the contributions and advantages of the proposed framework. Additionally, we provide a reproducible and comprehensible code for all experiments conducted in our study. For further information, please refer to the website <https://github.com/CsnowyLstar/HoGRC>. We believe that the revised paper offers a clearer and more readable version, including the writing, organization, figures, further

experiments, and discussions.

Comment 2

Reservoir computing is an excellent way to efficiently make data-driven forecasts of a dynamical system. However, isn't that exactly what is done in the paper [48] (they cite it, but not in a reasonable or substantive way. They cite it as a source for the famous Lorenz equations, but in fact, in [48], they perform exactly the same step of using dual RC forecasts to decide Granger-causality).

Reply. Thank you for your valuable comments. It is important to note that although the method described in the literature [48] (original manuscript) also employs RC for inferring the causal structure, it fundamentally differs from our framework.

Technically, they utilize the trained RC dynamics to estimate the Short Term Causal Dependence (STCD) metric $f_{ij} = G(\partial F_j(z)/\partial z_i)$. The metric f_{ij} characterizes the response of variable z_j to the perturbation of variable z_i . However, as the number of variables increases, the predictive performance of traditional RC steadily declines, rendering the STCD metric ineffective. This raises a crucial question of how to leverage potential interaction structures among these variables to facilitate better modeling of dynamic systems using RC. To this end, we propose the HoGRC framework. From this perspective, accurate prediction of dynamics is a crucial issue in our work, which is also the main difference between our work and the literature [48].

Moreover, their approach only considered the pairwise structure inference. However, recent studies show that the higher-order structures are vital to the emergence of complex dynamics [39], viz. diffusion [40], synchronization [41], and evolutionary processes [42]. To this end, we define the concept of higher-order neighbors in dynamical systems using simplicial complexes, and develop the HoGRC framework to simultaneously achieve scalable complex structures inference and accurate dynamics prediction.

To emphasize this difference, we briefly discuss this method in the Introduction section in a reasonable or substantive way.

Comment 3

Furthermore, what if the interaction between elements were not so simple as Lorenz, which is truly a simple system, but one where it is necessary to decide between direct or indirect influences of the sort $X \rightarrow Y \rightarrow Z$? It does not seem from my reading that this approach can decide if $X \rightarrow Z$ directly or only through Y . This is a classic and perhaps subtle issue, but it surprises me that the authors do not mention it since one of the co-authors is the lead author on several papers that is premised on exactly this issue, Sun, Jie, Dane Taylor, and Erik M. Bollt. "Causal network inference by optimal causation entropy." *SIAM Journal on Applied Dynamical Systems* 14.1 (2015): 73-106, But by entropy means. And several author papers by the co-author. The basic issue does not go away whether it is by entropy methods or by reservoir prediction. This is a fundamental issue.

Reply. Thank you for your thoughtful review and insightful question. Indeed, identifying indirect causality is a classical and fundamental issue in the task of structure inference. Accordingly, we add a brief discussion on this issue in the revised paper (See Section 2.6.3 of the main text and Section 1.3 of the Supplementary Material). For your convince, the summary is shown as follows.

In this study, we introduce a new computing paradigm called the higher-order RC, which aims to incorporate spatial information, particularly the higher-order structures, into the reservoir. However, the higher-order structure of a complex dynamical system is often unknown *a priori*. To address this, we integrate the concept of Granger causality (GC) into the higher-order RC to infer the underlying higher-order structures. These inferred structures are then integrated into the higher-order RC model to enhance its prediction performance. This iterative process is repeated until the model achieves optimal prediction accuracy. Consequently, we refer to this integrated model as the higher-order Granger RC (HoGRC) framework.

Throughout this process, GC inference and RC prediction are performed simultaneously and complement each other. Notably, the HoGRC does not require a precise learning of the system structure through GC. Instead, our framework focuses on optimizing the coupling structures to further maximize prediction accuracy. As a result, both direct and indirect causality can be inferred in the inference task. Despite this, our framework consistently and accurately infers the high-order structures in multiple experiments conducted in this study.

Figure R3: Detection of causal links from X and Y to Z .

We attribute the accuracy of our framework in the inferring tasks to several primary factors.

Firstly, many of the systems analyzed in our work display complex chaotic features, spurious causality can thereby lead to the rapid accumulation of errors, significantly reducing the predictive capacity of our framework in comparison to true causality.

Furthermore, the systems we study typically contain self-loops, as depicted by the node Y in Fig. R3(a). It allows us to accurately capture the direct neighbors rather than the indirect ones. In Fig. R3(a), the set of direct neighbors of node Z is $O_Z = \{X_3, Y\}$, while the set of direct neighbors of Y is $O_Y = \{X_1, X_2, Y\}$, where X_1, X_2, X_3 are higher-order nodes defined using the simplicial complex in our work, and Y and Z are the first-order nodes. Considering the indirect neighbors $\{X_3\} \cup O_Y$, it holds that $O_Z = \{X_3, Y\} \subseteq \{X_3\} \cup O_Y$. Since $Y \in O_Y$, we cannot exclude Y from the inference process. Otherwise, this may result in a poor prediction of Y solely based on $\{X_1, X_2\}$, as well as a poor prediction of Z based on $\{X_1, X_2, X_3\}$. Therefore, we ultimately eliminate neighbors X_1 and X_2 , which do not directly contribute to the prediction of Z , to obtain the true higher-order structure $O_Z = \{X_3, Y\}$.

When Y has no self-loop, the situation is different. As shown in Fig. R3(b), since $O_Y = \{X_1, X_2\}$, both $C_1 = \{X_1, X_2, X_3\}$ and $C_2 = \{X_3, Y\}$ can theoretically predict Z well. Therefore, the current HoGRC framework cannot distinguish this case. To address this issue, we extend our HoGRC framework to further identify the indirect causality via combining with the

existing methods. In particular, we propose two strategies as follows.

1. **Conditional Granger causality.** We define the conditional Granger causality from time-course X to Z conditional on time-course Y as,

$$F_{X \rightarrow Z|Y} = \ln \frac{\|e(Z_{\{Y\}})\|}{\|e(Z_{\{X,Y\}})\|},$$

where $\|e(Z_C)\|$ represents the average prediction error of node Z with the candidate neighbor set C (see Section 4.2 for details). According to this definition, when $F_{C_1 \rightarrow Z|C_2} \approx 0$, $\{X_1, X_2\}$ has no direct effect on Z , whereas when $F_{C_1 \rightarrow Z|C_2} > 0$, $\{X_1, X_2\}$ has a direct effect on Z .

2. **Further causal identification.** We can further consider the direct causal links between $\{X_1, X_2\}$ and $\{Y\}$ to eliminate the indirect causal cases, $\{X_1, X_2\} \rightarrow \{Y\} \rightarrow \{Z\}$ or $\{Y\} \rightarrow \{X_1, X_2\} \rightarrow \{Z\}$. This is because when both $\{X_1, X_2\}$ and $\{Y\}$ can independently predict Z , then only one of them is a direct cause, while the other is an indirect cause. More precisely, if one can accurately infer that $O_Y = \{X_1, X_2\}$, then Y is a direct neighbor of Z . In contrast, if $O_{X_1} = O_{X_2} = Y$, then X_1 and X_2 are direct neighbors of Z . To verify the effectiveness of these two strategies, we conducted additional experiments as described in Section 1.3 of the Supplementary Material. The experimental results demonstrate that our framework can be extended to identify indirect causality.

In order to enhance the completeness and rigor of the article, we have supplemented the aforementioned discussion as well as the method for identifying direct causality in the revised paper. We hope that the aforementioned discussion and additional modifications address the concerns you may have.

Comment 4

Also, the idea of using causation inference as associated with equation fitting exists in the literature, and here too, one of the co-authors is again a co-author on that work, so it is surprising that this current work is built as if this is a new idea. This is again substantial and not just something a new citation added will fix, since the central premise of this current paper already exists in the co-authors prior work. AlMomani, Abd AlRahman R., Jie Sun, and Erik Bollt. "How entropic regression beats the outliers problem in nonlinear system identification." *Chaos: An Interdisciplinary Journal of Nonlinear Science* 30.1 (2020), and again several author papers by the co-author, and others where the main premise that causation inference can be used to decide sparsity structure associated with modelling for efficient modelling. Rather than cite a half dozen papers for the last paragraph, and this one, since the author of this manuscript is on them, I will remind them, not just to cite, but to address the issue of deciding what is distinct other than RC is used, and as such how is it substantially different from [48].

Reply. Thank you for your valuable feedback. We apologize for the lack of clarity in some sections of our original manuscript, which may have led to your misunderstanding. To this end, we would like to provide clarifications on four aspects.

Firstly, our work mainly focuses on the integration of the structure information, specifically the higher-order structures, into the framework of RC, to improve the reconstruction and prediction of complex systems. This idea is primarily inspired by two papers in PRL (references [33] and [34]), where they only utilized pairwise structures to enhance the predictive capability of RC in high-dimensional systems. Therefore, our approach can be considered as a **new computing paradigm** of RC, as it incorporates the higher-order structures to achieve more accurate predictions, rather than solely utilizing RC for causal inference problems.

Secondly, as discussed in the **Reply** to **Comment 1**, in our HoGRC framework, GC inference and RC prediction are complementary. In other words, accurate prediction enhances the efficiency of GC inference, while high-quality inference facilitates the precise prediction of RC. On this basis, our HoGRC framework can compile the observed temporal data of systems as well as the higher-order structure information into the RC dynamics, enabling the achievement of structure inference and dynamics prediction simultaneously. From this perspective, our framework combines inference and prediction tasks, and provides a **powerful tool** for discovering the potential higher-order interactions and achieving more accurate

predictions from the observational data.

Thirdly, it is essential to highlight that previous works have explored the concept of utilizing causation inference for equation fitting. However, there are fundamental differences between these existing methods and our framework. Typically, these methods involve selecting a set of basis functions, such as polynomial or Fourier basis, to approximate the unknown system, and then employing the sparsity-promoting method, the information-theoretic regression technique to discover the governing equations. In contrast, our HoGRC framework does not rely on the use of any basis functions or delay-coordinate embedding technique. Additionally, we are the first to integrate the higher-order structures into the RC, which allows for more accurate dynamics prediction. By employing this RC with Granger causality, we can infer the higher-order structure of the underlying unknown system. This integration of higher-order structures sets our HoGRC framework apart from previous methods. Moreover, the HoGRC framework achieves the dynamics prediction in a node-level manner, exhibiting high scalability and efficiency. It is demonstrated through several experiments that our framework is much more robust and efficient for modeling complex systems compared to the simple equations fitted through inference.

Lastly, our framework differs significantly from the one proposed in reference [48]. For more detailed information, please refer to the **Reply** to **Comment 2** for details.

We hope that these discussions and additional clarifications adequately address your concerns.

Comment 5

Finally, and also importantly, whether fitting the equations directly without sparsity structure, or fitting by emphasizing sparsity structure as is done with SindY, or by the methods just described, its not a matter of this method got the right answer, especially for the simple problems illustrated here. It's a matter of efficiency, likely in terms of computational complexity. How well does this method do, and compare to the others, in terms of fitting quality in some measure of quality, with respect to growing data set size.

Reply. Thank you for your careful reading and valuable comments. It should be noted that

as mentioned in the **Reply** to **Comment 4**, our framework differs from previous methods, such as SindY, in that we do not rely on accurately inferring the dynamic equations based on a set of basis functions. When there are more complex interactions within the system, these methods have an increasing risk of producing an erroneous sparse model. Such incorrect identification of interactions may inevitably lead to catastrophic predictive performance, while a simple RC even without any structure information can often yield satisfactory results. Moreover, with the increase of system dimensions and the enhancement of nonlinearity, these basis function-based methods may struggle to fit the equation well and make accurate predictions (see **Section 2.4** on experiments of network dynamics for details). Additionally, our approach goes beyond previous work [48] (see the **Reply** to **Comment 2** for details), by introducing a novel computing paradigm for achieving higher-order structure inference and dynamics prediction simultaneously, instead of simply applying the RC to deal with these tasks.

In terms of the computational complexity, our framework exhibits remarkable efficiency, primarily due to the inherent efficiency of RC and the parallel prediction strategy at the node level. In addition, training the weight matrix W_{out} in the output layer poses a simple ridge regression problem. Notably, due to the nature of the node-level manner, the HoGRC framework allows for parallelization, thereby enhancing the efficiency in searching for higher-order structures and performing multi-step predictions.

Furthermore, in **Section 2.6.2**, we conduct a detailed analysis of the impact of varying dataset sizes on different methods. The results indicate that our method is capable of accurately capturing real dynamical mechanisms even with a smaller number of data points. In **Section 2.4**, we further analyze the effects of system dimensions and noise on different methods. Encouragingly, all of these analyses yield positive results on our framework.

We hope that the additional clarifications, the experimental findings, and the revised manuscript may adequately address your issues and concerns. Thank you very much again for your time and efforts.

To Reviewer #3's Report:

Comment 1

The authors present an original technique to analyse higher order complex networks by a promising combination of Granger causality with an extended kind of reservoir computing. They demonstrate the potential of their new method for various model systems of low and intermediate dimension as well as for the UK power grid. Compared to other methods theirs outperforms them clearly in the prediction and numbers of iteration steps. This is a very interesting and clearly written paper with much novelty. Therefore I recommend it for publication after a minor revision.

Reply. We sincerely appreciate your positive feedback and helpful comments. We have thoroughly considered your suggestions and addressed the concerns point by point in the following. To facilitate the identification of the changes, we have uploaded a revised version of the manuscript, where all modifications have been highlighted in red color.

Comment 2

The coupling strengths used are rather large (0.4 ... 0.5). What about smaller ones which are often typical in real systems? Is there a minimum coupling strength which they can retrieve?

Reply. Thank you for your careful reading and insightful question. Indeed, the coupling strength γ is an important parameter in our experiments. As mentioned in **Section 2.4**, a very small coupling strength implies a weak coupling effect on the dynamics, while sufficiently strong coupling tends to increase the predictability due to a high probability of synchronization occurrence. Therefore, in our experiments, we purposely selected a moderate level of coupling strength to increase the prediction difficulty.

To verify the aforementioned statement, we conducted additional experiments by varying the coupling strengths in the coupled Rössler system (CRoS). As depicted in Fig. R4(a), it is observed that the lower coupling strength does not significantly affect the predictive performance of our method. However, upon exceeding a certain threshold, the subsystems exhibit synchronization. The dynamics thereby becomes extremely predictable. Furthermore, Figures R4(b)-(d) demonstrate that the higher coupling strengths are associated with easier

Figure R4: Experimental results of the HoGRC method under different coupling strengths in CRoS. a The Performance of extrapolation prediction under different coupling strengths. b-d The structural inference of node B under different coupling strengths. Here, the value of γ is set to 0.01, 0.1, and 1, respectively.

coupled structure inference tasks. Specifically, Figure R4(d) provides clear evidence that the coupling neighbors of node B are $\{A, C, D\}$ (refer to Definition 3 in the manuscript for the detailed inference strategy). Conversely, when the coupling strength is low, as depicted in Fig. R4(b) ($\gamma = 0.01$), the difference in metrics $e(B)$ for various candidate structures decreases, thereby increasing the difficulty of the inference task. Nevertheless, as HoGRC is a delicate method by integrating the structures, one can still accurately identify the coupling structure by choosing a smaller value of ϵ_e in **Definition 3**. However, it is worth noting that this choice may inevitably reduce the robustness of the inference.

To enhance the comprehensiveness of the article, the aforementioned discussion has been added to Section 4.6 of the Supplementary Material.

Comment 3

I do not understand how many layers they have used (only n is given as the number of nodes in the reservoir). How many parameters to estimate for their algorithm?

Reply. Thank you for your valuable comments. In fact, the reservoir computing (RC), a special recurrent neural network (RNN), used in this work only has one hidden layer, i.e., the reservoir. Specifically, for the observational time series $x(t)$ generated from an unknown q -dimensional dynamical system, we embed it into a high-dimensional reservoir dynamics through a transformed signal, i.e. $W_{\text{in}}x(t)$, where W_{in} is called the input matrix of dimension $n \times q$. In particular, the n -dimensional reservoir dynamics $r(t)$ are designed as follows:

$$r(t) = (1 - l) \cdot r(t - 1) + l \cdot \tanh[Ar(t - 1) + W_{\text{in}}x(t) + b_r],$$

where l is the leaky rate, A is the weighted adjacency matrix of the reservoir network, and b_r is the bias term. Then, we decode the hidden state space $r(t)$ into the data space $x(t)$ via an output matrix W_{out} of dimension $q \times n$, yielding a predicted signal $\hat{x}(t) = W_{\text{out}}r(t)$. Typically, the matrices W_{in} and A in RC are randomly generated, with only the $q \times n$ parameters in W_{out} being trained using a closed form. In fact, the exceptional computational efficiency of our HoGRC framework primarily stems from the efficiency of the RC.

Comment 4

The model for power grids does NOT include losses which are very present in real ones (e.g. Nat. Comms. 11, s41467-020-14417-7 (2020)). This should be included in the model as well.

Reply. Many thanks for your valuable comments. We have thoroughly read the article you provided and agree that the introduction of losses in the Kuramoto model may be more realistic. As they pointed out, this loss term is ultimately treated as a phase lag α , which is often set as a constant value. It is important to note that introducing this constant may not influence the effectiveness of our framework.

In contrast to the pairwise interactions discussed in their research, we focus on the higher-

order Kuramoto model as mentioned in reference [53]. Consequently, the formulation of the higher-order model with losses becomes more complex. To maintain the concision of the main text and the consistency with the citation [53], we have included the content related to losses in Section 4.7 of the Supplementary Material. We hope these supplementary results adequately address your concerns.

Comment 5

In Fig. S6 the number of positive LEs should be given explicitly because it is almost impossible to estimate from the Fig.

Reply. Thank you for your careful reading and helpful comments. We have included the specific values for LEs in the description of Fig. S6 (Fig. S9 in the revised paper).

Finally, we would like to take this opportunity to express our sincere gratitude to all the reviewers for their insightful comments and valuable suggestions. Their constructive feedback does help us improve the overall quality of this work in terms of the writing, readability, experiments, and methodology. We believe that the revised version of this article has significantly enhanced not only in terms of the framework refinement and further analysis but also in the experimental aspect. We hope that the individual responses to each reviewer have effectively addressed their concerns, and we are more than happy to answer any further questions to make this work more readable and valuable. We would appreciate it if the reviewers could acknowledge our efforts and further contributions to this revised work.

Reviewers' comments:

Reviewer #1 (Remarks to the Author):

Thank you for the answer. I think that some answers are interesting.

Concerning the code, thank you also. Unfortunately, only 2 codes are running on my machine (main_CL63.py and main_L96.py). I think you really need to make an effort to provide good codes that run without any difficulty. Moreover, you need to explain them and to explain what they are intended to. Currently, it is true you provide them but I have the feeling, this is a work that was too quickly done.

Your paper is slightly better now, but for me there are still many parts that are not explained correctly.

In the first version of your paper, I imagined that your work could be used to make any prediction of time series.
Now I understand it is clearly not possible.

I still think you are going directly into the results without providing a clear context of your work.

So if someone is a specialist in your domain they will understand. If someone like me understands well RC especially for time series prediction, I think your work is not clear enough...

So I still think you need to clarify many details. You consider that readers have your background. You need to clearly explain what your work is intended for and what it is not. Give more understandable examples.

Reviewer #3 (Remarks to the Author):

The revised version is much better and all my points are clearly answered. Therefore, I recommend acceptance of this ms.

Reviewer #5 (Remarks to the Author):

The manuscript by Li et al. "Higher-order Granger reservoir..." presents a new paradigm for reservoir computing (RC) that employs the ideas of Granger causality and higher-order interactions to enhance the reservoir computing performance. The algorithm is intended for predicting data obtained from dynamical systems of potentially high dimension. The proposed method includes not only a classical RC inference, but also an additional interactive analysis of a structure of a dynamical model. With an inferred structure, the reservoir is iteratively updated.

The amount of numerical results and material presented is impressive. However, despite of the amount of material and many illustrations, I missed a clear and transparent explanation of the method. I also have some other critical observations, which I mention below. In summary, I do not recommend this paper for publication. Below, I focus on the criticism to justify my recommendation. On the other hand, I am sure that the paper contains interesting

results, and maybe it can be submitted to another journal, or even resubmitted to NatComm, but after a reworking.

More specific comments:

- The explanation of the method is not transparent and clear. For example, Sec. 2.1, which is the first place where one would expect some more direct explanation, introduces some notations, but then refers to the "Methods" section. However, the "Methods" also section contains unclear and undefined notations. Consider, for example, Proposition 1. Since it is formulated as a proposition, I expect a rigorous statement and a proof. However, the notation of calligraphic H is used without definition, and it is really unclear what it is. It is not enough to say that these are "function families". GE is not defined either. And these are just two examples, there are others.
- The title of the paper contains the word "scalable". I may have missed it, but I have not seen any substantiation for the use of this word.
- On page 3, the authors use the term "nodes" without introducing a network or graph. I don't see any explanation at the beginning of the paper why a network is considered at the input.
- I don't understand why the emphasis is on higher-order interactions? What is the difference to the case of a simple graph?
- The resulting reservoir is not only sparse, but also contains blocks of disconnected components (see e.g. the fig. 1 (d)). This seems counterintuitive and should be explained. This point is not a criticism, but a question.
- The lower part of Fig. 1 is unclear.
- The definition of the Lyapunov time as λt_1 is unclear and possibly wrong.
- Why is VPT used in the case of high-dimensional systems and VPT otherwise?
- The Lorenz63 system on page 8 does not include higher-order interactions.
- On page 10, the authors claim "Since the performance of the HoGRC framework is independent of network scale and is node-level based...". In my opinion, this is not substantiated in the paper.
- Also another less strong claim on page 10, "As shown in Fig. 3a, the dynamics reconstruction and prediction using the HoGRC framework perform well", refers to fig. 3a, which does not show much information.

Reply to Reviewers' reports

To Reviewer #1's Report:

We sincerely appreciate the time and efforts you devoted to reviewing our work. Your comprehensive, valuable, and insightful comments helped us significantly improve the overall quality of this work. We acknowledge that the initial round of revisions may still lack clear methodological clarity and readability, and we sincerely apologize for any confusion or misunderstanding that may have arisen. Accordingly, we extensively revised the manuscript to enhance its readability and tried our best to address all of your specific concerns thoroughly and carefully. To facilitate the revision process, we have uploaded a revised version of our manuscript with modifications highlighted in red for your convenience.

Comment 1

Concerning the code, thank you also. Unfortunately, only 2 codes are running on my machine (main_CL63.py and main_L96.py). I think you really need to make an effort to provide good codes that run without any difficulty. Moreover, you need to explain them and to explain what they are intended to. Currently, it is true you provide them but I have the feeling, this is a work that was too quickly done.

Reply. Many thanks for your valuable comments. Indeed, we acknowledge that our initial code had the readability issue. As a result, we have carefully refined the code and ensured that all experiments are executable.

To enhance the readability of our code, we have thoroughly refined the code comments and “readme” file. Additionally, we have supplemented two illustrative examples, available in the files “An_Example_for_Task_1.py” and “An_Example_for_Task_2.py”. The first example produces the results corresponding to Fig. 2a of the main text, and the second yields the outcomes represented in Figs. 3a and 3c of the main text. The source code for all experiments can be accessed at the following GitHub repository: <https://github.com/CsnowyLstar/HoGRC>, which can also be tested online via the Code Ocean capsule (**preferable**) at: <https://codeocean.com/capsule/3561870/tree/v1>. It should be noted that for code execution on personal devices, it is advised to download the source code from GitHub, given that running the code from Code Ocean locally may have path issues.

In addition, regarding the additional experiments described in the main text and supplementary materials, the configurations within the code need to be adjusted according to the specific experimental requirements. In the following, we use the coupled Lorenz63 (CL63) system as a concrete example to explain the implementation process of structure inference and dynamics prediction. As described in the file “main_CL63.py”, our code can be divided into five parts.

- (1) The first part is **hyperparameter setting**, where all relevant hyperparameters are stored in the variable “args”.
- (2) The second part is **data generation**, invoking the file “dataset/Data_CL.py” to generate experimental data of the CL63 system. And this data is then stored in the “dataset/data” folder.
- (3) The third part is **the configuration of higher-order structures**, which are used as additional inputs for our higher-order RC.
- (4) The fourth part is the **model training**. During the model import, we mainly considered three models: RC, PRC, and HoGRC. These models correspond to the files named “Model_RC.py”, “Model_PRC.py”, and “Model_HoGRC.py”, which are located in the “models” directory. It should be noted that we employ the high-order structure from part (3) when calling the HoGRC model and use ridge regression to train the output layer parameters W_{out} .
- (5) The last part is the **model testing**. In testing, we evaluate the one-step and multi-step predictive errors of the updated higher-order RC model. For the task of structure inference, the one-step predictive error serves as the metric for updating higher-order structures in part (3), where Algorithm 1 is used in the manuscript. After sufficient iterations of the structure inference, the algorithm will converge to an optimal higher-order structure (**Task I**). Then, the optimal model can be used to accurately achieve the multi-step dynamics prediction (**Task II**).

Comment 2

Your paper is slightly better now, but for me there are still many parts that are not explained correctly.

In the first version of your paper, I imagined that your work could be used to make any prediction of time series. Now I understand it is clearly not possible.

Reply. Thank you for your comment. We sincerely apologize for this misunderstanding probably due to the heavy materials of our previous resubmission. Significantly, the ultimate goal of our work is to make accurate dynamics predictions by incorporating the (inferred) higher-order structure.

In this revision, we have provided more details and enhanced the clarity and transparency of our framework. Several important changes are listed as follows:

1. **Figure 1** (see the main text). We meticulously redrew the schematic diagram of our framework.
2. **Box 1** (see the main text). We further provided the main steps on how our framework specifically works.
3. **Reorganization of the manuscript.** To provide a clear context of our work before we go directly into the results, we introduce preliminaries on classical reservoir computing and some key definitions used in our manuscript, and then present our framework.

Comment 3

I still think you are going directly into the results without providing a clear context of your work.

So if someone is a specialist in your domain they will understand. If someone like me understands well RC, especially for time series prediction, I think your work is not clear enough...

So I still think you need to clarify many details. You consider that readers have your background. You need to clearly explain what your work is intended for and what it is not. Give more understandable examples.

Reply. Thank you for your careful reading and valuable comments. As mentioned in the **Reply** to **Comment 1**, we have supplemented two examples in our source code. Moreover, we hope our important changes in this revision, along with the source code, effectively address your concerns. Thank you again for your time and efforts.

To Reviewer #3's Report:

Comment 1

The revised version is much better and all my points are clearly answered. Therefore, I recommend acceptance of this ms.

Reply. We sincerely appreciate the time and efforts made by the reviewer in thoroughly reviewing our work, and thank you very much for the very positive recommendation of publication of our revised manuscript.

To Reviewer #5's Report:

We sincerely appreciate the time and efforts made by the reviewer in thoroughly reviewing our work. Accordingly, we have carefully considered your valuable comments and suggestions, and addressed the concerns from this expert point by point. This has significantly contributed to improving the overall readability of the revised manuscript. We have also refined the experiments and the source code to make the contributions of our work even clearer. Our revised manuscript has been uploaded, with the changes clearly highlighted in red for clarity.

Comment 1

The explanation of the method is not transparent and clear.

Reply. Thank you for your comment. We sincerely apologize for the unclear explanation of the method, probably due to the heavy materials of our previous resubmission. In this revision, we have provided more details and enhanced the clarity and transparency of our framework. Several important changes are listed as follows:

1. **Figure 1** (see the main text). We meticulously redrew the schematic diagram of our framework.
2. **Box 1** (see the main text). We further provided the main steps on how our framework specifically works.
3. **Reorganization of the manuscript.** To provide a clear context of our work before we go directly into the results, we introduce preliminaries on classical reservoir computing and some key definitions used in our manuscript, and then present our framework.

Comment 2

For example, Sec. 2.1, which is the first place where one would expect some more direct explanation, introduces some notations, but then refers to the "Methods" section. However, the "Methods" also section contains unclear and undefined notations. Consider, for example, Proposition 1. Since it is formulated as a proposition, I expect a rigorous statement and a proof. However, the notation of calligraphic \mathcal{H} is used without definition, and it is really unclear what it is. It is not enough to say that these are "function families". $\mathcal{G}\mathcal{E}$ is not defined either. And these are just two examples, there are others.

Reply. Many thanks for your valuable comment. We sincerely apologize for the cross-referencing issue as well as the unclear and undefined notations in our manuscript. In this revision, as mentioned in the **Reply** to **Comment 1**, we reorganized the manuscript to provide a clear presentation of our work and we further substantially addressed the problem of unclear and undefined notations in our manuscript, including the above two examples raised by this expert.

Comment 3

The title of the paper contains the word "scalable". I may have missed it, but I have not seen any substantiation for the use of this word.

Reply. Thank you for your valuable comments. Notably, our framework fully inherits the scalability or the parallel merit of the existing work [1]. More precisely, different from the traditional RC, we use a specific higher-order RC \mathcal{R}_u to model a given node u , thereby requiring a smaller reservoir size n or resulting in a lightweight model. Moreover, since all lightweight reservoirs \mathcal{R}_u are independently trained, our framework can be efficiently processed in parallel, which in turn makes our framework scalable to higher-dimensional systems.

The scalability of our framework can be further understood through the following three aspects:

1. **A simple illustrated example.** Our HoGRC framework is a node-level approach capable of incorporating the structural information to make dynamics predictions for all nodes in parallel (see the Fig. 1 and the Box. 1 in the main text). Here, for your convenience, we provide a simple example that as shown in Fig. 1, for the state variable or node y in the Lorenz system, we input the 3-dimensional data $\mathbf{x} = [x, y, z]^T$ and the higher-order neighbors \mathcal{S}_y of node y into a specific higher-order RC \mathcal{R}_y , and output the one-step prediction of node \hat{y} . Notably, the process is the same for all nodes, leading to the scalability of our framework.
2. **Experiment on the CRoS system.** We further conduct experiments on the CRoS system (consisting of m coupled subsystems and each subsystem being a 3-d system) with different scales, small to intermediate sizes ($15 \leq 3m \leq 150$), and different kinds

Fig. 1: The process of the one-step prediction at the node level for each state variable or node. Here, the node y is chosen as an illustrative example, yielding a specific higher-order RC \mathcal{R}_y as shown in the figure.

of coupled networks (ER & BA), as detailed in Section 2.6.2 of the main text. The results are depicted in Fig. 6, which demonstrates that our method is scalable and can be applied to larger-scale systems. Moreover, our framework exhibits superior performance over baseline methods in various coupled network structures.

- 3. UK power grid system.** Our framework is versatile and can be used in various complex systems, further demonstrating its scalability in practical applications. Specifically, we conducted experiments on ten distinct systems, including a large-scale UK power grid system, as presented in both the main text and supplementary material.

Comment 4

On page 3, the authors use the term "nodes" without introducing a network or graph. I don't see any explanation at the beginning of the paper why a network is considered at the input.

Reply. Thank you for your careful reading and valuable comment. Accordingly, in the revision, we introduced the concept of hypergraph as well as the "nodes" term (see Section 2.1.2, the paragraph below Definition 2) and added an explanation at the beginning of presenting our framework why a network is considered at the input (see Section 2.2.1).

Comment 5

I don't understand why the emphasis is on higher-order interactions? What is the difference to the case of a simple graph?

Fig. 2: The difference between the simple graph and the hypergraph.

Reply. Thank you for your valuable comment. Actually, the introduction of higher-order structures is one of the most fundamental contributions of our work. Recently, the PRC framework [1] integrated pairwise structures to predict dynamics in complex systems. However, they cannot reveal the higher-order structures, which can be seen as a more precise representation of the complex interactions in complex dynamical systems.

For your convenience, we provide an illustrated example to highlight the difference between the simple graph (pairwise structures) and hypergraph (higher-order structures). As shown in Fig. 2, for the case of the simple Lorenz63 system, the concept of the simple graph corresponds to the pairwise structure depicted in the left of Fig. 2. On the other hand, the higher-order structure depicted in the right of Fig. 2 extends the neighbors of nodes to higher-order neighbors described by the simplicial complexes. For further details regarding higher-order structures, please refer to Section 2.1.2 in the main text.

Comment 6

The resulting reservoir is not only sparse, but also contains blocks of disconnected components (see e.g. the fig. 1 (d)). This seems counterintuitive and should be explained. This point is not a criticism, but a question.

Reply. Thank you for your question. In the traditional RC method, the input matrix \mathbf{W}_{in} and the adjacency matrix \mathbf{A} are typically randomly initialized, without leveraging any spatial structure information about the dynamical system. However, in our HoGRC framework, we aim to embed the structure information, particularly the higher-order structure information, into the RC framework.

Specifically, different from the traditional RC, we use a specific higher-order RC \mathcal{R}_u to

model a given node u and meticulously incorporate its higher-order structure information into the \mathcal{R}_u . Moreover, all reservoirs \mathcal{R}_u are independently trained in our framework. Therefore, this is consistent with your observation that the resulting reservoir (the previous revision) is not only sparse, but also contains blocks of disconnected components. In this revision, to clearly show the node-level property, we dissect our framework into a specific higher-order RC \mathcal{R}_u for each node u as shown the Fig. 1 in the main text. For more specific details regarding the embedding process, please refer to Section 2.2.1 of the revised manuscript.

Comment 7

The lower part of Fig. 1 is unclear.

Reply. Thanks for your comment. The redraw Fig. 1 in the main text is now more clear.

Comment 8

The definition of the Lyapunov time as λt_1 is unclear and possibly wrong.

Reply. Many thanks for your careful reading and helpful advice. Indeed, it is incorrect to represent “Lyapunov time” using λt_1 . To standardize the evaluation metrics, we use the simple metric VPS, as defined in Section 2.2.3, throughout the manuscript.

Comment 9

Why is VPT used in the case of high-dimensional systems and VPT otherwise?

Reply. Many thanks for your comment. To standardize the evaluation metrics, we use the simple metric VPS, as defined in Section 2.2.3, throughout the manuscript.

Comment 10

The Lorenz63 system on page 8 does not include higher-order interactions.

Reply. Thank you for your comment. As shown in Fig. 2, the Lorenz63 system indeed has higher-order interactions. For example, the node z has a higher-order neighbor $\{x, y\}$.

Comment 11

On page 10, the authors claim "Since the performance of the HoGRC framework is independent of network scale and is node-level based...". In my opinion, this is not substantiated in the paper.

Reply. Thank you for your careful reading and helpful comment. The above claim is not accurate, and we thereby modified it as "Since the HoGRC framework is a node-level based...". Please refer to the **Reply** to **Comment 3** for the node level or scalability of our framework.

Comment 12

Also another less strong claim on page 10, "As shown in Fig. 3a, the dynamics reconstruction and prediction using the HoGRC framework perform well", refers to Fig. 3a, which does not show much information.

Reply. Thank you for your instructive comment. Actually, the above claim is not accurate, and we thereby modified it as "As shown in Fig. 3a, the trajectory predicted by our HoGRC framework closely matches the true trajectory of the FHNS system."

Figure 3a (in the main text) illustrates the prediction performance of our method in the FHNS system, where the solid blue line represents the actual system data, and the red dashed line represents the multi-step predictive result. To further enhance the clarity and visual presentation of the figure, we utilize blue solid dots and red hollow circles to represent the locations of actual data points and predicted data points, respectively. The predictions of our method match the real data very well, which supports its prediction ability.

Finally, we would like to express our sincere gratitude to the four esteemed reviewers, including the previous reviewers #1 #2, #3 as well as the additional reviewer #5, for their profound insights, constructive comments, and meaningful suggestions, which have been instrumental in enhancing the quality of this manuscript. We hope that the individual responses to each reviewer have effectively addressed their concerns, and we are more than happy to answer any further questions to make this work more valuable and comprehensive. We would appreciate it if the reviewers could acknowledge our efforts and further contributions to this revised work. Many thanks again for the time and efforts invested by the reviewers and editor in reviewing our manuscript.

References

- [1] Keshav Srinivasan, Nolan Coble, Joy Hamlin, Thomas Antonsen, Edward Ott, and Michelle Girvan. Parallel machine learning for forecasting the dynamics of complex networks. Physical Review Letters, 128(16):164101, 2022.

REVIEWER COMMENTS

Reviewer #1 (Remarks to the Author):

Why do you use two functions f ? One as a nonlinear vector field and one as a scalar function.

In order to clearly understand the difference between equation 2 and equation 5. I think you must give an example (which is clear).

Currently, both equations are very similar and because you consider that everyone has your knowledge, and because you give absolutely no example, it is very difficult to understand this paper. In each new revision, so many things change that it shows your paper is missing something very important: clarity.

In the new github version. My previous comments are still the same.

Your code is not running. Here is the output.

I still do not understand why you are not able to make a python code which can be simply executed. All the other codes I see on github are good except yours...

```
python -m An_Example_for_Task_2
edges
Data generated successfully!
#####RC#####
initialize_internal_weights
train...
initialize_input_weights
Traceback (most recent call last):
File "<frozen runpy>", line 198, in _run_module_as_main
File "<frozen runpy>", line 88, in _run_code
File "/home/couturie/ajeter/HoGRC/An_Example_for_Task_2.py", line 132, in <module>
experiment.train()
File "/home/couturie/ajeter/HoGRC/models/Model_RC.py", line 151, in train
joblib.dump(readout, './models/model/readout.pkl')
File "/home/couturie/tools/anaconda3/lib/python3.11/site-packages/joblib/numpy_pickle.py",
line 552, in dump
with open(filename, 'wb') as f:
^^^^^^^^^^^^^^^^^^^^
FileNotFoundError: [Errno 2] No such file or directory: './models/model/readout.pkl'
```

```
python -m An_Example_for_Task_1
Data generated successfully!
#####HoGRC-7-7-7#####
initialize_internal_weights
train...
Traceback (most recent call last):
```

```
File "<frozen runpy>", line 198, in _run_module_as_main
File "<frozen runpy>", line 88, in _run_code
File "/home/couturie/ajeter/HoGRC/An_Example_for_Task_1.py", line 114, in <module>
experiment.train()
File "/home/couturie/ajeter/HoGRC/models/Model_HoGRC.py", line 155, in train
joblib.dump(readout,'./models/model/readout'+str(ni)+'_'+str(Vi)+'.pkl')
File "/home/couturie/tools/anaconda3/lib/python3.11/site-packages/joblib/numpy_pickle.py",
line 552, in dump
with open(filename, 'wb') as f:
^^^^^^^^^^^^^^^^^^^^^^^^^^^^^^^^
FileNotFoundError: [Errno 2] No such file or directory: './models/model/readout0_0.pkl'
```

Consequently my opinion is the following. Your paper is maybe interesting but you don't have the ability to explain it clearly. You are not able to make a code that is running. So I am really disappointed by this paper. I am not saying the work is bad. I am just saying that currently I am not able to understand it because the description is not clear and because the code is not running....

Reviewer #1 (Remarks on code availability):

I used the github code and the code has some errors....

Reviewer #5 (Remarks to the Author):

The authors give very clear and thorough answers to all my comments. In my opinion, the presentation is now much clearer and more rigorous. As for the significance of the results: I think that the use of higher order interactions gives additional insight into the problem of possible optimal structures of artificial neural networks. This additional insight can have a significant impact on the development of ML methods for time series prediction.

In principle, I could recommend the manuscript for publication. However, there is a problem with the code, see my remark on the code.

Reviewer #5 (Remarks on code availability):

The code is executable in Code Ocean, but I had problems running it on my PC, especially because the data is stored in the root directory, e.g. as `plt.savefig("/results/output1.png")`. Is it possible to improve this, e.g. by specifying relative paths?

Reply to the Reviewers' reports

To Reviewer #1's Report:

We sincerely appreciate the time and efforts you devoted to reviewing our work. Your comprehensive, insightful, and valuable feedback, particularly regarding readability and clarity, has significantly enhanced the overall quality of our manuscript. To enhance the clarity of our work for readers with limited background knowledge, we have included simple examples to explain our method. In addition, we have thoroughly addressed issues of code non-executability to facilitate the replication of all experiments in our work. To facilitate the revision process, we have uploaded a revised version of our manuscript with modifications highlighted in red for your convenience.

Comment 1

Why do you use two functions f ? One as a nonlinear vector field and one as a scalar function.

Reply. Many thanks for your careful reading and valuable comment. The boldface letter \mathbf{f} denotes a vector-valued function in Eq. (1) of the main text, which can be expressed as

$$\dot{\mathbf{x}}(t) = \mathbf{f}[\mathbf{x}(t)] = (f_1[\mathbf{x}(t)], f_2[\mathbf{x}(t)], \dots, f_N[\mathbf{x}(t)])^\top,$$

where f_1, f_2, \dots, f_N are scalar functions. To avoid any confusion, the scalar function “ f ” mentioned in Definition 1 has been renamed to “ g ”, which can be regarded as a placeholder for any scalar function, such as $f_i, i = 1, 2, \dots, N$.

Comment 2

In order to clearly understand the difference between equation 2 and equation 5. I think you must give an example (which is clear). Currently, both equations are very similar and because you consider that everyone has your knowledge, and because you give absolutely no example, it is very difficult to understand this paper. In each new revision, so many things change that it shows your paper is missing something very important: clarity.

Reply. Many thanks for your helpful comment and constructive suggestion. For your convenience, we consider the Lorenz63 system as a simple example to illustrate the difference between higher-order RC and the traditional RC, and we also visually display this difference in Fig. 1. The primary distinction between our method and traditional RC lies in the

Fig. 1: The difference of the RC with hidden dynamics of Equation (2) and the HoGRC with hidden dynamics of Equation (5).

exploitation of the higher-order structural information to achieve the node-level prediction, with the details as follows.

- **RC.** In the traditional RC framework, both the input matrix \mathbf{W}_{in} and the adjacency matrix \mathbf{A} are randomly initialized without incorporating any structural information as visually shown in Fig. 1. Actually, only a *single* RC with the readout layer is used for the prediction of the whole system state $\mathbf{x} = (x, y, z)$.
- **Higher-order RC.** We use the Lorenz63 system for illustration. Actually, we have a total of *three* sub-RC networks with higher-order structures because there are *three* state variables, viz. $x, y, z \in V$ for this system. Different sub-RC has different input and adjacency matrices, notably incorporating high-order structures. Particularly, we consider a specific node (state variable) $u = z$ in the Lorenz63 system, as shown in Fig. 1. We write out

$$\dot{z} = f_3(x, y, z) = -\beta z + xy = g(x, y, z) = g_1(z) + g_2(x, y),$$

where $g_1(z) \triangleq -\beta z$, $g_2(x, y) \triangleq xy$, and $D_z \triangleq 2$. Consequently, according to Def-

initions 1 & 2 in the main text, the set of the higher-order neighbors of node z is $\mathcal{S}_z = \{\mathbf{s}_{z,1}, \mathbf{s}_{z,2}\} = \{\{z\}, \{x, y\}\}$.

As mentioned above, the set of the higher-order neighbors of node z is $\{\{z\}, \{x, y\}\}$ with $D_z = 2$. Thus, we obtain $\tilde{\mathbf{W}}_{\text{in},z} = [\psi^\top(z), \psi^\top[(x, y)]]^\top$ according to the notations set in Equation (6) of the main text, where the third column of $\psi^\top(z)$ and the first and the second columns of $\psi^\top[(x, y)]$ are the random sparse submatrices, and the remaining parts are zero submatrices. Moreover, we obtain $\tilde{\mathbf{A}}_z = \text{diag}\{\varphi(z), \varphi[(x, y)]\}$, which is a block diagonal matrix comprising two random sparse submatrices. Therefore, we encode this higher-order structural information into the input matrix $\tilde{\mathbf{W}}_{\text{in},z}$ and the adjacency matrix $\tilde{\mathbf{A}}_z$, yielding the Equation (5).

Finally, the following module is the readout layer through the matrix $\tilde{\mathbf{W}}_{\text{out},z}$, employed for the prediction of node z . And due to our approach operates at the node level, we can execute predictions for all nodes $\mathbf{x} = (x, y, z)$ in parallel.

In the current revised manuscript, below Eq. (5), we accordingly demonstrate the difference between the traditional RC and the higher-order RC. Additionally, we include this specific illustrative example in the Section 1.3 of Supplementary Material.

We note that the higher-order structures of the system are often unknown or partially known, thus our whole framework includes the higher-order structure inference (the Fig. 1c in the main text) and the multi-step dynamics prediction (the Fig. 1d in the main text). We therefore kindly recommend referring to the Fig. 1 and the Box 1 in the main text to gain a conceptual understanding of the workflow of our framework. Building on this foundation, we employed the simple Lorenz63 system to elucidate our HoGRC framework more illustratively, with the following specific modifications:

- In the final paragraph of Section 2.1.2 in the main text, we delineate the definitions of higher-order neighbors and structures using the Lorenz63 system.
- We refine Fig. 3a and present its accompanying description about how to infer the higher-order RC in the first example of the Lorenz63 system in Section 2.3 of the main text. We also provided a **demo** in the Github repository (<https://github.com/CsnowyLstar/HoGRC>, “Simple_example.gif”) to more concretely illustrate the process of structure inference using the simple example of node z in the Lorenz63 system.

Comment 3

In the new github version. My previous comments are still the same. Your code is not running. Here is the output. I still do not understand why you are not able to make a python code which can be simply executed. All the other codes I see on github are good except yours...

Reply. Many thanks for your valuable comment. We sincerely apologize for our oversight that led to the inability to run the code from GitHub locally. This issue arose because the empty folder intended for storing model files was automatically deleted upon upload to GitHub. To rectify this, we have thoroughly addressed all potential issues, and can assure that the latest code will successfully replicate all experiments presented in this paper as long as you follow our steps.

To facilitate replication of our experiments on personal computers by readers, we provide the complete executable code on GitHub with python version $\geq 3.9.7$, accessible at the following repository: <https://github.com/CsnowyLstar/HoGRC> (recommended for use on personal computers). Meanwhile, we provide an executable Code Ocean capsule accessible at: <https://codeocean.com/capsule/7064569/tree/v1> (recommended for use online).

Again, we would like to apologize for any lack of clarity in our presentation and any issues with our code. Finally, we are immensely grateful for the time you have generously devoted, the considerable effort you have invested, and the invaluable comments and suggestions you have provided. Your feedback has been greatly appreciated.

To Reviewer #5's Report:

Comment 1

The authors give very clear and thorough answers to all my comments. In my opinion, the presentation is now much clearer and more rigorous. As for the significance of the results: I think that the use of higher order interactions gives additional insight into the problem of possible optimal structures of artificial neural networks. This additional insight can have a significant impact on the development of ML methods for time series prediction.

In principle, I could recommend the manuscript for publication.

Reply. We sincerely appreciate the time and efforts made by the reviewer in thoroughly reviewing our work, and thank you very much for the very positive recommendation of publication of our revised manuscript.

Comment 2

However, there is a problem with the code, see my remark on the code. The code is executable in Code Ocean, but I had problems running it on my PC, especially because the data is stored in the root directory, e.g. as `plt.savefig("/results/output1.png")` Is it possible to improve this, e.g. by specifying relative paths?

Reply. Thank you for your helpful comment. In fact, code downloaded from Code Ocean may encounter path issues when executed on personal computers, as intermediate and result files on Code Ocean must be stored in the root directory, including folder “/results”.

To facilitate replication of our experiments on personal computers by readers, we provide the complete executable code on GitHub with python version $\geq 3.9.7$, accessible at the following repository: <https://github.com/CsnowyLstar/HoGRC> (recommended for use on personal computers). Meanwhile, we have refined the code on Code Ocean to enable local execution, and the updated Code Ocean capsule is accessible at <https://codeocean.com/capsule/7064569/tree/v1> (recommended for use online).

REVIEWERS' COMMENTS

Reviewer #1 (Remarks to the Author):

Now the explanations are finally clearer. I am satisfied with the current version of the paper. The code is also good. I have tried it and it runs perfectly. I am sure it will help researchers to be able to use your work.

Reviewer #5 (Remarks to the Author):

The authors addressed my comments satisfactorily.

Reply to Reviewers' reports

To Reviewer #1's Report:

Comment 1

Now the explanations are finally clearer. I am satisfied with the current version of the paper. The code is also good. I have tried it and it runs perfectly. I am sure it will help researchers to be able to use your work.

Reply. We sincerely appreciate the time and efforts made by the reviewer in thoroughly reviewing our work, and thank you very much for the very positive recommendation of publication of our revised manuscript.

To Reviewer #5's Report:

Comment 1

The authors addressed my comments satisfactorily.

Reply. We sincerely appreciate the time and effort the reviewer has invested in thoroughly reviewing our work, and we are extremely thankful for the overall positive evaluation of our work.